

# Effects of plant traits on the regulation of water cycle processes in the Amazon Basin

Kien Nguyen [1] and Maria J. Santos [1]

[1]Department of Geography, University of Zurich, Switzerland

**Correspondence:** Kien Nguyen (kien.nguyen@geo.uzh.ch)

**Abstract.** Plants play a key role in the soil-plant-atmosphere-climate hydrological continuum as they depend on water for their persistence and in turn affect water exchange processes. Changes in plant composition may affect these relationships through induced changes in cover, composition and functionality; however, detailed understanding on how feedbacks that involve plant traits develop are still seldom included in observational, experimental and modeling studies. To address this gap, here we make
use of datasets derived from Earth Observation and models to examine the effect of plant traits on water cycle processes in the Amazon Basin. We used quantile regression to examine how plant traits (Specific Leaf Area (SLA), Leaf Dry Matter Content (LDMC), Leaf Phosphorus Content (LPC) and Leaf Nitrogen Content (LNC)), respond to parameters related to regulation of atmospheric water content (Evapotranspiration (ET), Potential Evapotranspiration (PET), Vapour Pressure Deficit (VPD)), land surface temperature (Land Surface Temperature (LST) day and night)), and soil moisture content (Soil Moisture (SM))
along their range of values. We found that SLA had the strongest relationships with parameters involved in the regulation of atmospheric water content and land surface temperature, but weak relationships with regulation of soil moisture content, for the Amazon basin and its sub-basins. Plant traits show even stronger relationships at the 5th and the 95th quantiles; this is particularly strong at low values of ET and PET and high values of VPD and LST. The associations remain strong and localised in some particular sub-basins. Our results highlight the role of plant traits in mediating hydrological processes, which are not
yet included in current models. Further, the results suggest that if climate change induces shifts in water cycle parameters to more extreme values, the functional response of plants may exacerbate these effects and affect the resilience of the Amazon forest.

## 1 Introduction

Water cycle processes are increasingly affected by climate changes (Tabari, 2020; Konapala et al., 2020). Yet, plant functions
that regulate water exchange, transport and storage (Matthews, 2006) could play a role in mediating these effects. Plants are fundamental in the uptake, transport and exchange of water from the soil to the atmosphere (Jackson et al., 2000; Caldwell et al., 1998) driving soil-plant-atmosphere interactions (Katul et al., 2012). Alterations in these interactions may lead to changes in the hydrological regime and in the provision of precipitable water, alter resilience to hydrological shocks (Keys et al., 2019; Zemp et al., 2014; O'Connor et al., 2021; Zemp et al., 2017; Satyamurty et al., 2012), and in turn affect the general biosphere
distribution (Tang et al., 2014). As such, plants controls on hydrological flows are fundamental to the response of ecosystems



to droughts, floods and other water redistribution processes such as moisture recycling (Salati et al., 1979; Keys et al., 2017; O'Connor et al., 2021; Van Der Ent et al., 2014). Vegetation cover, composition and function affect hydrological processes over space (te Wierik et al., 2021) and time (Caballero et al., 2022), yet the effects of plant traits on these processes have only recently come forward in the literature (Funk et al., 2017). Plant traits, i.e. morphological, anatomical, physiological, biochemical
and phenological characteristics of plants at individual, community or ecosystem levels (Kattge et al., 2011), and trait-based approaches (Garnier and Navas, 2012) are well established in ecology (Green et al., 2022) to understand plant relationships with their environment. As such, expanding this framework to examine the role of plants on hydrological processes could prove useful, as evidenced in the growing literature on, for example, hydraulic traits (Anderegg, 2015; Anderegg et al., 2019).

Plant functional traits might influence single or multiple hydrological processes (Matheny et al., 2017). For example, hy-
draulic traits mediate ecosystem responses and resilience to drought (Anderegg et al., 2018), how variation in bark thickness provides resistance to fire (Staver et al., 2020), and how plant trait diversity improves ecosystem function (Yan et al., 2023) and increases ecosystem resilience (Sakschewski et al., 2016). Plants exert controls on water fluxes through their role in transpiration (Kool et al., 2014; Christoffersen et al., 2014), photosynthesis (Gu et al., 2003), rainfall interception (Magliano et al., 2022; Van Dijk and Bruijnzeel, 2001), and root water uptake (Aroca et al., 2012; O'connor et al., 2019; Ehleringer and Dawson,
1992). For example, transpiration alone accounts for 61% of evapotranspiration (ET) globally, and in tropical regions it can contribute as much as 70% (Schlesinger and Jasechko, 2014). These rates in the fractional contribution of transpiration to ET can be affected by changes in vegetation cover and composition. For example, (Wang et al., 2010) reported a 22% increase in ET associated with an increase in woody plant cover from 25% to 100%, which varied during the growing season (Ashktorab et al., 1994). Further, the more abundant and larger the leaves are, the higher rate of transpiration (Kool et al., 2014; Christof-
fersen et al., 2014). This is because stomatal regulation controls 80-90% of the water exchange to the atmosphere (Verma and Verma, 2007), and this process is affected by wind and solar radiance, temperature, relative humidity, the structure of the canopy, the number of stomata in a leaf surface, and the percentage of stomata opening (Zhao et al., 2013; Ward, 1971; Feng et al., 2020). Interception also greatly impacts the water cycle (Miralles et al., 2010; Savenije, 2004), as it may account for 10%-40% of the total precipitation (Crockford and Richardson, 2000; Gash et al., 1980). Larger leaves likely intercept more
water, making this water available for latent heat exchanges and contributing moisture to the atmosphere (Magliano et al., 2022; Van Dijk and Bruijnzeel, 2001). Photosynthesis not only depends on water availability but also responds to water exchange at the leaf and canopy levels (Gu et al., 2003) as well as water uptake from the soil. For example, leaf water exchange, stomata density, specific leaf area, and xylem pressure at 50% conductivity, hydraulic safety margin, and the water potential at 50% loss of stem hydraulic conductivity mediate land–atmosphere feedbacks that dictate tolerance of plants to droughts and
their regulating effect on water exchange (Powell et al., 2017; Anderegg et al., 2019). Root depth controls soil water uptake, and such process is mediated by soil water holding capacity and affects other processes in the soil-plant-atmosphere-climate continuum (O'connor et al., 2019). Yet plant physiological responses to changes in atmospheric water content, land surface temperature, and soil water content remain an underexplored component in the water cycle, and better understanding of the direction and the magnitude of plant trait effects and how they vary spatially is required.



Direct links between plant traits and hydrological processes may be hard to assess because of the direct and indirect effects
of single or multiple traits on these processes and their variation within and among species (Van Bodegom et al., 2012), the
scales at which plant-water exchange may occur from stomata to whole ecosystems (Aleixo et al., 2019), and the geographical
variation in the relationship with varying environmental conditions (Aleixo et al., 2019). First, plant traits either respond to or
are an effect of given environmental conditions (Wolf et al., 2022; Chapin, 2003), i.e., the former dictates community responses
to environmental change while the later refers to the effect of that change on ecosystem processes such as those characteristic
of the water cycle. For example, hydraulic traits could be categorized as response traits to water availability but also could be
categorized as effect traits if they mediate water exchange processes. Second, plant traits do not operate in isolation, i.e. occur
as a set of co-functioning traits – trait syndromes (Pan et al., 2021). Some traits are activated upon the enactment of a previous
trait, e.g., root characteristics and depth determine water available to plants, which upon transport to the canopy may activate
stomatal conductivity and determine leaf water exchange (Mencuccini et al., 2019). In a trait syndrome, some traits may exert
a positive effect on plant-water relations, while others might have a negative effect, and the net effect may vary along a gradient
of environmental conditions. For example, the soil–plant continuum breaks if forced to transport water beyond its capacity
(Hultine et al., 2020), and leaf cooling and hydraulic properties exhibit strong trade-offs along a gradient of aridity (Blasini
et al., 2022). While synergies and trade-offs in trait functioning led to coordinated co-evolution of traits (Sanchez-Martinez
et al.), in some cases these might be mismatched or show no responses to water stress, especially in regions that have not yet
experienced this type of stress (Signori-Müller et al., 2021). Third, net effects of plant traits on hydrological processes may also
be determined by intra- and interspecific variation in trait values as these vary by species, plant communities and ecosystems
(Kattge et al., 2020, 2011). Fourth, there is a variety of scales at which plant-water exchange occurs, from leaf stomata to
canopy and ecosystems, and these scales are hard to capture in both observations and models. A variety of models have been
devised to predict the relationship between trait and water processes at the stomata level (e.g. (Lu et al., 2020)), leaf level
(e.g. (Collatz et al., 1992)), individual level (e.g. Joshi et al. 2022), canopy level (e.g. (Mirfenderesgi et al., 2016)), and global
level (e.g. moisture recycling tracking model (Tuinenburg and Staal, 2020)). The integration of such models is often difficult,
and generally only a set of processes are included in these models (Baudena et al., 2015). Observational studies and statistical
models together with AI-based models, now complement the range of process-based models used to study the effects of traits
on hydrological processes, as these can capture scale effects (e.g. remote sensing and field measurements (Liu et al., 2021)).
Experiments seldom occur at larger scales, thus data from plot level and certain geographies is often used and extrapolated
to large scale processes. Some plant-water processes are not conducive for experimental work given the scale at which they
operate (e.g. moisture recycling), therefore limiting the ability to connect models and processes where desirable. Fifth, the
effect of traits on water cycle processes is likely to differ across regions and varying environmental conditions. For example,
plant hydraulic traits vary over gradients of water stress (Hultine et al., 2020; Anderegg, 2015) and water logging (Blasini et al.,
2022). These environmental gradients lead to variability in the trait values themselves, and in turn may be affected by plant
traits' role in mediating hydraulic processes.

Here we examine the effect of plant traits on three processes that depict soil-plant-atmosphere-climate interactions, namely
regulation of atmospheric water content, regulation of land surface temperature, and regulation of soil water content. We





hypothesize that plant traits will respond to changes in these processes along a gradient of environmental conditions, and to test this hypothesis we use quantile regression to relate a set of remote-sensing based estimates of plant trait distributions with water process parameters over the Amazon basin. We chose the Amazon because the persistence of the forest largely relies on its water cycle, with 25% of its precipitation being contributed by regional ET (Eltahir and Bras, 1994). Vegetation cover in the Amazon not only contributes to ET, but also to interception and formation of condensation (Hasler and Avissar, 2007;

Xu et al., 2019; Casagrande et al., 2021; Zheng and Jia, 2020), the maintenance of soil moisture dynamics (Laio et al., 2001), reduction of soil erosion and floods (Durán Zuazo and Rodríguez Pleguezuelo, 2008), modulates water run-off to streams and oceans (Nagase and Dunnett, 2012; Blanusa and Hadley, 2019), and hydraulic regulation on water flux and water status (Deng et al., 2017). As such, we ask: (i) is there a relation between plant traits and water cycle parameters in the Amazon Basin?, (ii) if so, is this relationship sensitive to the extreme values of water cycle parameters?, and (iii) do the relationships change for the

sub-basins within the Amazon? We expect that by examining plant traits that respond to gradients of water processes, we can advance our understanding of eco-hydrological processes that are mediated by the biosphere. We expect this information to be also useful to guide conservation strategies that ensure the resilience of the Amazon's hydrological and ecological processes that are fundamental to maintain this very important biome and its livelihoods.

## 2   Methods

### 2.1   Study area

The Amazon Basin is a vast region of 6.3 million $km^2$, encompassing the Amazon river and its distributary channels, bounded between coordinates $((-80.5°, -48.5°), (6°, -20.5°)$, Figure 2). The basin includes seven sub-basins, the Amazonas, Madeira, Negro, Solimões, Tapajós, Trombetas and Xingu. The majority of the basin is covered by rainforest (5.5 million $km^2$), of which 60% occurs within Brazil, 13% in Peru, 10% in Colombia and the rest to countries including Venezuela, Ecuador, Bolivia,

Guyana, Suriname and French Guiana. Being the largest rainforest on Earth, the Amazon holds a great importance as it is one of the regions with highest biodiversity globally, as well as being fundamental to regulating the Earth's climate (Gatti et al., 2021; Flores et al., 2024).

### 2.2   Plant traits and water cycle parameters data

We chose to analyze a set of traits that we hypothesize to have an effect on the regulation of atmospheric water content,

regulation of land surface temperature, and regulation of soil water content (Figure 1). We expect that larger and heavier leaves likely have higher number of stomata (Wang et al., 2019), leading to higher transpiration and more interception (Magliano et al., 2022; Van Dijk and Bruijnzeel, 2001); hence, high Specific Leaf Area (SLA) and low Leaf Dry Matter Content (LDMC) likely lead to high Potential and Actual Evapotranspiration (PET and ET). With high ET, evaporative water demand is likely to decrease at the canopy level (Massmann et al., 2019), as well as resulting in evaporative cooling (Chakraborty et al., 2021).

Thus, we expect that high SLA and low LDMC would result in low Vapour Pressure Deficit (VPD) and day and night time




Land Surface Temperatures (LST). With low VPD, there is a lower demand for water for evaporative processes which in turn reduces the demand for soil moisture (SM) (Wang et al., 2021), thus we would expect that high SLA and low LDMC would lead to high SM. Finally, since plants in tropical regions tend to be phosphorus limited (Turner et al., 2018) and recent evidence has emerged specifically for the Amazon (Cunha et al., 2022), this might also affect evaporative fluxes, cooling and demands

for soil water. Thus, we would expect low Leaf Phosphorus Content (LPC) to be associated with low ET and PET, and high VPD and LST. Since nitrogen is also limiting in some cases (Figueiredo et al., 2019), then we would also expect that low Leaf Nitrogen Content (LNC) would lead to low ET and PET and high VPD and LST. As such, the net effect of trait interactions with the water cycle processes will likely depend on the balance of the effects of SLA and LDMC and that of LPC and LNC, and these effects will likely vary along the gradients of the water parameters.

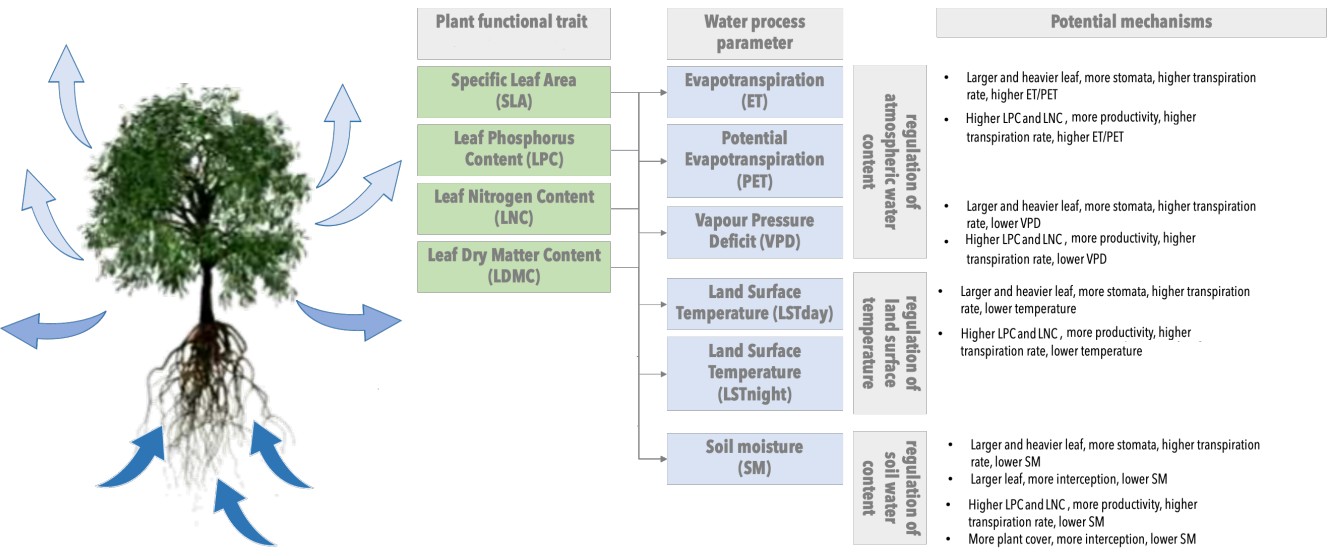

**Figure 1.** Conceptual diagram of the relationships between plant traits and water process parameters

Several global spatially explicit trait distribution maps have been produced by modeling the relationship between in-situ trait measurements and satellite remote sensing and other auxiliary data (Dechant et al., 2023). The best predictions of plant functional traits at global scale include those of (Moreno-Martínez et al., 2018), who computed trait distributions for SLA, LDMC, LNC and LPC over a time period from 2000–2010 and at 1km and 3km spatial resolution using a random forest model, of which used the 3km version in our analyses (Table 1, Figure 1).

We also chose a set of parameters related to the three water cycle processes we focus on this paper. For regulation of atmospheric water content, we chose to analyze Evapotranspiration (ET), Potential Evapotranspiration (PET) and Vapour Pressure Deficit (VPD). For regulation of land surface temperature, we use Land Surface Temperature (LST) at day and night (LSTday and LSTnight), and for regulation of soil water content, we chose a measure of Soil Moisture (SM). For ET and PET, we used the Moderate Resolution Imaging Spectroradiometer (MODIS) MOD16A3 dataset (Mu et al., 2013), at 500m





**Figure 2.** (A) Study area location, (B) Sub-basins of the Amazon basin, (C) average ET and average PET based on MOD16A3 data (Running et al., 2021), average VPD based on TerraClimate data (Abatzoglou et al., 2018), LST based on MOD11C3 data (Wan et al., 2015) and SM based on (Guevara et al., 2021) between 2001 and 2010, and (D) average SLA, LDMC, LNC and LPC (Moreno-Martínez et al., 2018))

resolution and yearly between 2001 and 2010 to match the timeframe of the trait data (Table 1). For VPD, we used the TerraClimate dataset (Abatzoglou et al., 2018), at 4km, monthly, and again between 2001 and 2010. For LST, we used the MODIS MOD11C3 dataset (Wan et al., 2015) at $0.05°$ spatial resolution and with monthly temporal resolution also between 2001 and 2010. Finally, for SM, we used a global soil moisture dataset (Guevara et al., 2021), at monthly frequency for the same time period, at $0.25°$ resolution.

It is also important to note that, within the model developed by (Moreno-Martínez et al., 2018), the most influential climatological variables used for trait prediction do not intersect with the ones that we used in our study. Notably, the influential variables identified by (Moreno-Martínez et al., 2018) predominantly relate to precipitation (BIO12-17 which correspond to





| Trait/water cycle parameters | Unit | Spatial Resolution | Temporal Resolution | Source |
|---|---|---|---|---|
| LDMC | g/g | 3km | 10-year average | Moreno-Martínez et al. (2018) |
| SLA | mm2/mg | 3km | 10-year average | Moreno-Martínez et al. (2018) |
| LPC | mg/g | 3km | 10-year average | Moreno-Martínez et al. (2018) |
| LNC | mg/g | 3km | 10-year average | Moreno-Martínez et al. (2018) |
| ET | mm/year | 1km | Yearly | Mu et al. (2013) |
| PET | mm/year | 1km | Yearly | Mu et al. (2013) |
| VPD | kPa | 4km | Monthly | Abatzoglou et al. (2018) |
| LST (Day/Night) | Kelvin | $0.05°$ | Monthly | Wan et al. (2015) |
| SM | m3/m3 | $0.25°$ | Monthly | Guevara et al. (2021) |

**Table 1.** Data sources

Annual precipitation, Precipitation of driest month, Precipitation seasonality, Precipitation of wettest quarter, Precipitation of driest quarter in Table A.1), with temperature-related variables such as maximum annual temperature (BIO1), isothermality (BIO3) and Mean temperature of warmest quarter (BIO10) playing a lesser role. The one exception is LST, which was listed as an input variable for the trait model (Table 2 in (Moreno-Martínez et al., 2018)), yet it does not contribute significantly to the mapping of the traits (Table D1 in (Moreno-Martínez et al., 2018)).

### 2.2.1 Data pre-processing

We chose the base spatial resolution of $0.05°$ because it is the resolution of the NDVI data used to masking of vegetated areas (MOD13 dataset by Didan (2021)) and the closest to the 3-km resolution of the plant trait data (Moreno-Martínez et al., 2018). We then either downsampled or upsampled the datasets with different spatial resolutions using a cubic resampling method to match the base resolution. Gaps in the resampled datasets were filled by linear interpolation. We calculated the 10-year mean and standard deviation for ET, PET, VPD, LSTday, LSTnight and SM. We also selected only vegetated pixels for the analysis by defining a NDVI threshold greater or equal to 0.3, as this enables to separate vegetation and non-vegetation areas (Hashimoto et al., 2021; Fragal et al., 2016). This resulted in 191,932 pixels for further analysis.

### 2.2.2 Data analysis

We used quantile regression to examine the association of plant traits and each of the parameters for the three processes we are examining. Quantile regression has been frequently used (Brennan et al., 2015; Good and Caylor, 2011; Liu et al., 2013; Cade et al., 2005), since it examines how responses of a given variables vary across the range of values that it may take (Cade and Noon, 2003). The rational is that while there might be a weak or non-existent relationship between the mean of the response variable and that of the predictors, stronger relationships might occur with other segments of the response variable distribution ((Cade and Noon, 2003). In our case it helps identify changes in the shape of the relationship between plant traits and water





process parameters across their range of values. We conducted quantile regressions for each of the trait-water cycle parameter pair (48 pairs in total). For each relationship, we evaluated the relationship at the 5th quantile (extreme low values), the 50th (median value), and the 95th quantile (extreme high values). The quantile regression was performed using the *statsmodel* package (v0.13.2: www.statsmodels.org) in Python 3.6.13. We calculated a pseudo R-square (Koenker and Machado, 1999) to represent the goodness-of-fit of the models. We did the analysis for the whole Amazon and separately for each of the sub-basins. The boundaries of the sub-basins were taken from the HydroBASINS database (Lehner and Grill, 2013).

## 3  Results

### 3.1  Amazon Basin:

We found significant relationships between plant traits with both the regulation of atmospheric water and land surface temperature, but not with the regulation of soil water content (Figures 3, 4 and 5). We found that SLA, LDMC, and LPC, but not LNC, have significant relationships with most water cycle parameters at the 50th quantile but mostly with low regression coefficients. Model slope sign and coefficients become higher at the 95th quantile, while regressions either become not significant or switch the sign of the regression slope at the 5th quantile (3). The explanatory power of the quantile regression models tends to be higher for the 95th quantile and lower for the 5th quantile models, except for ET_mean (Figure 3).

### 3.1.1  Regulation of atmospheric water content

Generally, SLA and LDMC were associated with atmospheric water content parameters with a more dominant role at extreme values, while both LPC and LNC showed almost no influence. We found seven relationships between plant traits and parameters related to the regulation of atmospheric water content. Two of these relationships occur between SLA, LPC, and VPD_sd at the 50th quantile, while the other five emerge with the extreme quantiles. The models varied in their explanatory power, with the highest pseudo-$R^2 \approx 0.18$. (Table: A9). The relationships became stronger at high values and decoupled at low values of plant traits (Figure 3). At the 50th quantile, SLA and LDMC showed opposite relationships with atmospheric water content parameters, with SLA negatively related with ET but positively with both PET and VPD and the opposite for LDMC (Figures 4). Further, LPC showed mostly negative relationships with the median of all parameters, while LNC showed mostly positive relationships. At the extreme values, SLA showed stronger positive relationships with PET_mean and VPD_mean at high values but no relationship at low values, while we found the opposite for ET. For LDMC, we found that at high values the negative relationships become more negative with PET and VPD, while the relationship with ET_mean becomes not significant, and we observe the opposite at low values. Finally, for both LPC and LNC the extreme values maintain the same direction of the relationship (Figure 4).



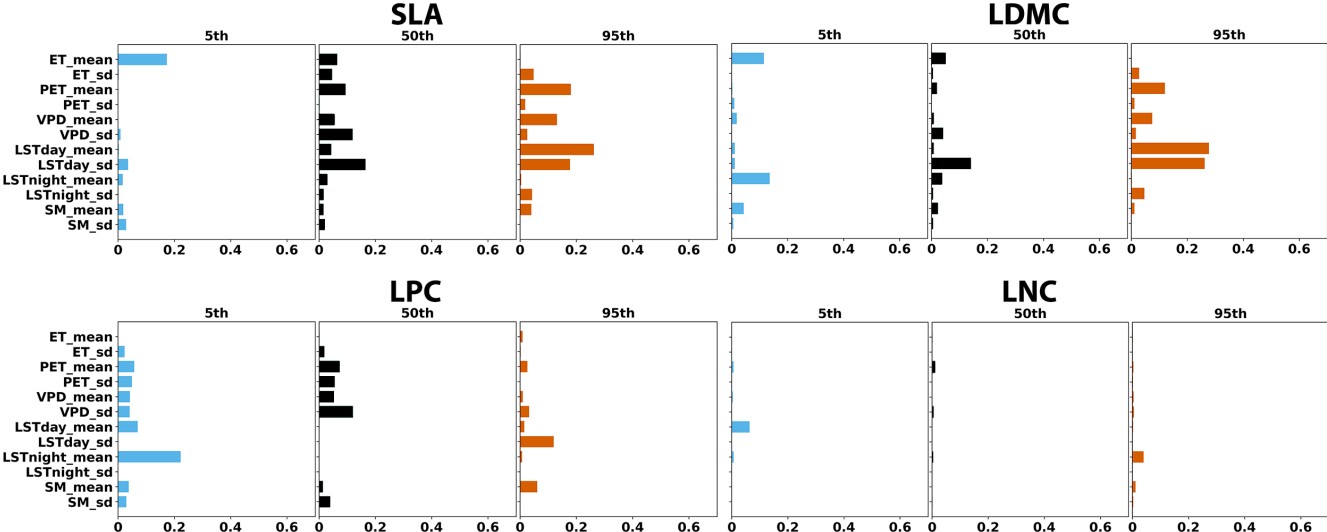

**Figure 3.** Quantile regression of the effect of plant traits on water cycle parameters. Graphs depict model performance at the 5th, 50th and 95th quantiles, i.e. the pseudo $R^2$ of each of the models significant at $p < 0.0001$.

### 3.1.2 Regulation of land surface temperature

We found stronger associations of plant traits with land surface temperature parameters than with atmospheric water content, and these relationships vary at extreme values. The slope of the relationships increases at low values of LST, compared to the median and extreme high values. However, the explanatory power is notably stronger at extreme high values, with

four strong correlations (see Figure 3). SLA and LDMC show opposite patterns at both median and extreme high values of LSTday_sd, with SLA having a positive relationship ($R^2 = 0.166, \beta^* = 0.451$) and LDMC having a negative relationship ($R^2 = 0.142, \beta^* = -0.582$). This contrast between SLA and LDMC also occurs at the extreme high values of LSTday_mean (SLA: $R^2 = 0.263, \beta^* = 0.553$; LDMC: $R^2 = 0.278, \beta^* = -0.706$). Further, only LPC and LDMC are associated with extreme low values of LSTnight_mean, (LPC: $R^2 = 0.224, \beta^* = -1.270$; LDMC: $R^2 = 0.137, \beta^* = 0.930$).

### 210   3.1.3 Regulation of soil moisture content

Trait effects on the regulation of soil moisture content were mostly not significant. We also found no differences in model performance at either extreme of soil moisture values. The highest $R^2$ value that we could observe is 0.062 between LPC and the extreme high values of SM_mean.




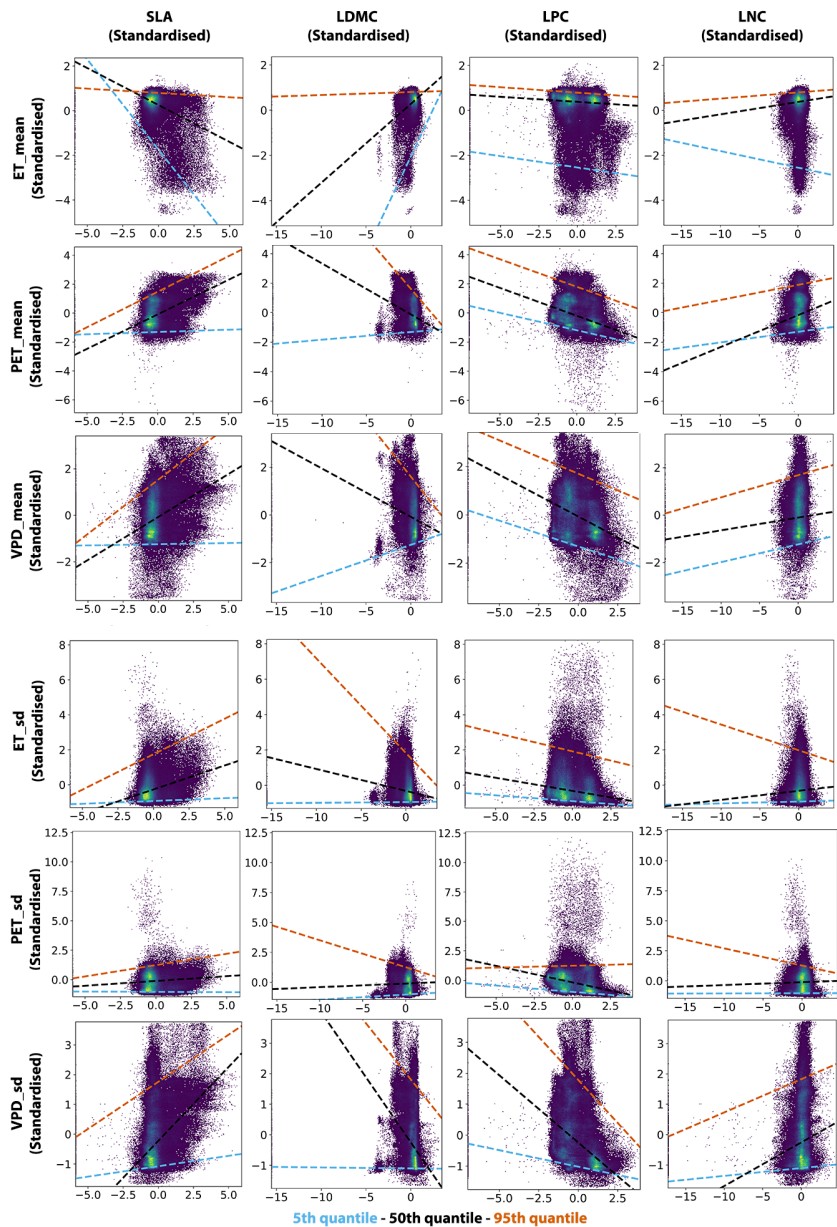

**Figure 4.** Quantile regressions for the relation between plant traits and environmental parameters related to regulation of atmospheric water content for the Amazon Basin.





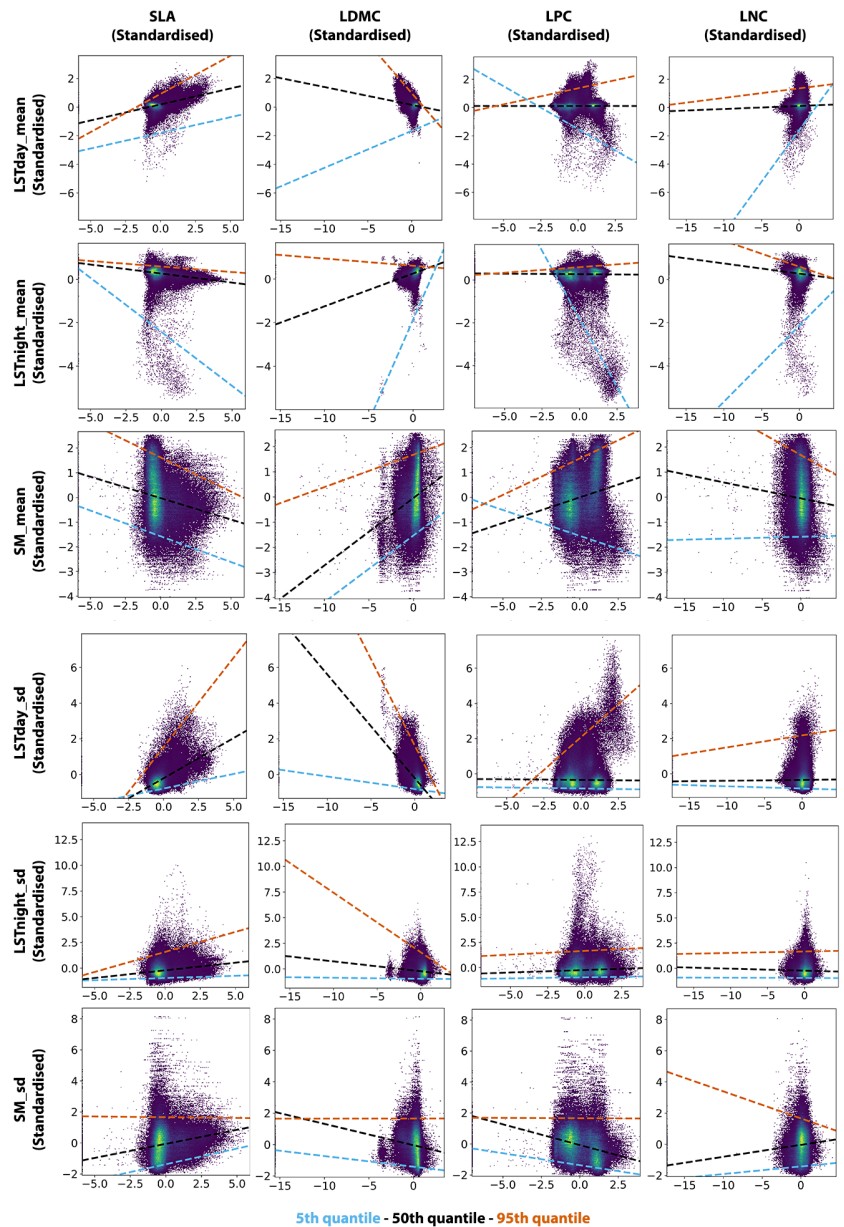

**Figure 5.** Quantile regressions for the relation between plant traits and environmental parameters related to regulation of surface temperature and soil moisture for the Amazon Basin.

## 3.2 Sub-basins:

At the sub-basin level we found three key trends: 1. Some associations between plant traits and water parameters were strong at the whole basin and become stronger at the sub-basin; 2. Some associations become weaker; and 3. Some associations that were weak at the whole-basin level become strong at the sub-basin level. We detail the foundings in the next sub-sections.





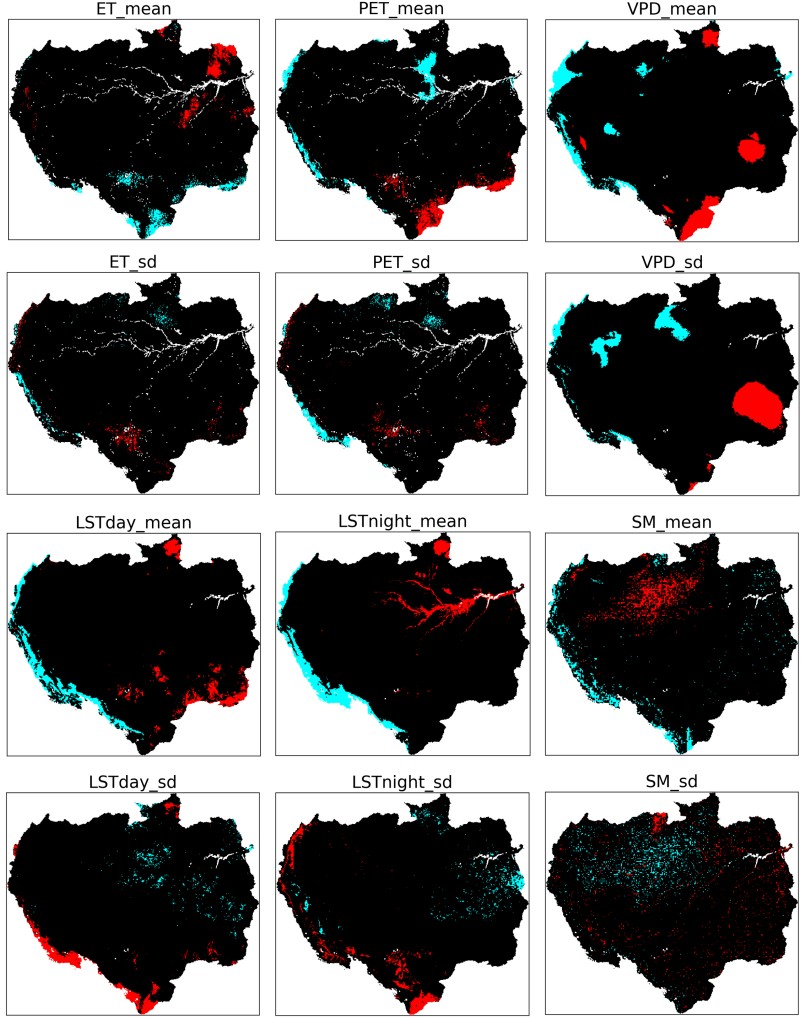

**Figure 6.** Location of extreme values of water cycle parameters (extreme low - 5th quantile: blue, extreme high - 95th quantile: red)

### 3.2.1 Regulation of atmospheric water content

For the associations that are strong the whole-basin level, we found intensification located at the Amazonas, Negro, Solimoes, Tapajos and Xingu sub-basins. Along with the intensification in some associations, other associations decrease in explanatory power and/or $\beta^*$, and none exhibit high explanatory power with a reversed direction of $\beta^*$ of the whole basin. In specific for the intensifying associations, we found that at extreme low values of ET, SLA model explanatory power increases for the Negro ($R^2 = 0.303, \beta^* = -1.068$) and Xingu sub-basins ($R^2 = 0.179, \beta^* = -0.748$). In addition, also for extreme low values of ET_mean, LDMC effects intensify in five sub-basins: Amazonas ($R^2 = 0.161, \beta^* = 1.4697$), Negro ($R^2 = 0.301, \beta^* = 0.95$), Solimões ($R^2 = 0.177, \beta^* = 0.702$), Tapajós ($R^2 = 0.174, \beta^* = 0.842$) and Xingu ($R^2 = 0.197, \beta^* = 0.941$). The intensifica-





tion related to SLA and LDMC also occurs at the extreme high values of PET_mean in Negro (SLA: $R^2 = 0.300, \beta^* = 0.877$, LDMC: $R^2 = 0.339, \beta^* = -1.039$). For the median values of VPD_sd, we observed its strengthened association with SLA in Negro ($R^2 = 0.229, \beta^* = 0.737$) and with LPC in Solimoes ($R^2 = 0.153, \beta^* = -0.524$)

For the associations that emerge only at the sub-basin level, the majority occur in the Negro and Tapajós sub-basins (13 relationships), and one in Trombetas and in Amazonas sub-basins. We also found new relationships with LPC: in Tapajós, where LPC becomes associated with extremely low values of ET_mean ($R^2 = 0.157, \beta^* = -0.744$) and the median value of PET_mean ($R^2 = 0.174, \beta^* = 0.530$); and in Trombetas, where LPC becomes associated with extremely low values of VPD_sd ($R^2 = 0.223, \beta^* = -0.626$). Additionally, new associations emerge between SLA and LDMC and extreme high values of PET_sd in Tapajós, and ET_sd, VPD_mean, VPD_sd in Negro; and other three cases associated with median values of VPD_mean and VPD_sd in Negro, and PET_mean in Amazonas.

### 3.2.2 Regulation of land surface temperature

We found that strong relationships between traits and regulation of land surface temperature remain at the sub-basins level. Notably, LDMC and SLA are associated with extreme high values LSTday_mean and LSTday_sd show in all sub-basins (Appendix A). In addition, associations between LDMC and SLA with LSTday_sd also remain strong at the sub-basin level. At the extreme low values of water cycle parameters, LSTnight_mean stays strong but decreases in explanatory power in Madeira and Solimoes, meanwhile LSTnight_mean decouples with LDMC.

Among associations that were weak at whole-basin but become strong at some sub-basins, we observe three foundings. First, LNC shows stronger associations at the extreme low values of LSTday_mean in Madeira, and extreme high values of LSTnight_mean in Amazonas and Trombetas. Second, LPC shows stronger associations with median values of LSTday_mean in Tapajos and Xingu, LSTday_sd in Tapajos, and LSTnight_mean in Xingu. Third, SLA and LDMC also show associations at the median values of LSTday_mean for Amazonas and Tapajos sub-basins, and LSTnight_mean for Amazonas and Xingu.

### 3.2.3 Regulation of soil moisture content

Similar to the whole-basin level, we found no trait effects on the regulation of soil moisture content at the sub-basin level. The highest $R^2$ values are observed in Tapajós (0.082 between SLA and extreme high values of SM_mean) and Solimões sub-basins (0.072 between LDMC and median values of SM_mean).

## 4  Discussion

In our study, we examined whether plant traits affect water related processes for the whole Amazon basin and its sub-basins. We found that at the Amazon basin scale, plant traits have strong effects on both the regulation of atmospheric water content and land surface temperature but not on the regulation of soil moisture content. These relationships generally became stronger at extreme values of water process parameters. More specifically, SLA and LDMC exerted the greatest influence on all water processes, with some contribution of LPC to the regulation of atmospheric water content and land surface temperature. We





also found generally consistent results for both the entire Amazon basin and its sub-basins, yet the effect became stronger and localised at the sub-basin level.

## 4.1 Plant traits and regulation of atmospheric water content

Plants with larger leaves likely intercept more water, making this water available to the atmosphere (Liu et al., 2019); yet, we found a negative effect of SLA on ET_mean while positive with PET_mean, VPD_mean and VPD_sd. These results could be due to the method and data we used. It could be that the model estimates of SLA (Moreno-Martínez et al., 2018) are not sufficiently accurate. Further, given the vertical stratification of SLA in the Amazon forest, lower at the top of the canopy and almost double at the bottom of the canopy (McWilliam et al., 1993), this could have an effect in what is measurable from
remote sensing. However, (Moreno-Martínez et al., 2018) reports a mean error of 0.01 and a root mean square error (RMSE) of 3.13 for the SLA models, and that SLA has lower values and the lowest RMSE for broad-leaf evergreen vegetation in comparison to other traits in tropical areas, which suggests that the modeled SLA values are reasonable. Further, the approach by (Moreno-Martínez et al., 2018) was deemed one of the best approaches to estimate traits in comparison with other global approaches (Dechant et al., 2023). This is in line with foundings from previous studies, which showed that SLA is generally
well estimated using remote sensing data, and tends to be more stable than the estimates of, for example, LNC and LPC (Asner and Martin, 2008). In addition, the predicted SLA values from (Moreno-Martínez et al., 2018) achieve higher correlation with those expected from process-based models based on theoretical optimality of plant function (Dong et al., 2023), and perform even better than other traits. It could also be that the negative relationship between SLA and ET_mean could emerge because we are using averages over a ten year period capturing some of effects of deforestation that could have affected the values
of SLA in specific areas. We do found that the relationship is most exclusively restricted to grids with extreme low values of ET_mean, in the Amazonas and Madeira sub-basins, where the majority of deforestation occurred (Acre - 3.2%, Rondônia - 12%, Beni - 3% (Potapov et al., 2022)). As primary forest was lost, degraded or replaced by secondary forest or other vegetation types, SLA could have decreased. Deforestation has been reported to significantly decrease ET (Davin and Noblet-Ducoudré, 2010; Devaraju et al., 2015) in particular in the Amazon (Baker and Spracklen, 2019; Heerspink et al., 2020). Mechanistically,
we know that old trees tend to have higher dry matter content and smaller leaf area compared to young trees (Lohbeck et al., 2013, 2015), and old leaves could have lower gas exchange capacity, which would result in lower ET values. Further, the leaves of Amazonian trees show a trade-off between tissue toughness with leaf size (Poorter et al., 2018), which could constrain SLA values and how they relate with ET. A meta-analysis conducted by (Niinemets et al., 1999) suggests that SLA may not have a consistent effect of photosynthesis as leaf thickness and leaf density showed opposing effects, and therefore could explain our
results for gas exchange. Finally, there is much local variation in traits that could result in the overall negative relationship for the whole Amazon. Our models were stronger at the sub-basin level, almost doubling their explanatory power thus suggesting that further analysis is needed to understand what drives the relationships between SLA and ET within the Amazon.

We found a negative relationship between LPC and PET_mean, VPD_mean and VPD_sd. The negative relationship with PET_mean could be attributed to the well established phosphorous limitation on photosynthesis of tropical forests (Mercado
et al., 2011), which would result in ET not meeting its potential. The negative relationship of LPC with VPD is aligned with the





reduction of VPD when there is more stomatal conductance. Higher LPC suggests that there is less limitation by phosphorous (Walker et al., 2014), thus higher photosynthetic activity.

## 4.2    Plant traits and regulation of land surface temperature

We found strong positive associations of SLA with LST at both basin and sub-basin scales, which align with the negative
relationship between SLA and ET. With high SLA yet low ET there is less evaporative cooling, hence higher LST as a result. Similarly, at the sub-basin scale we also found positive associations between SLA and extreme high values of LSTday_mean, while only one sub-basin maintains the relationship at extreme low values of LSTday_mean. As mentioned above, the relationship and effects of SLA on water cycle processes are still under debate, as such its relation with land surface temperature deserves more attention.

We also found that LPC is negatively related with LST. This could mean that the release of phosphorous limitation due to high LPC would result in more water exchange through the stomata, which in turn could contribute to evaporative cooling and decreases in land surface temperature. Limitation in phosphorus has been reported to affect plant adaptation strategies, particularly for the Amazon Basin (Quesada et al., 2012). Yet, at the sub-basin level, LPC switches the direction of the effect for a large fraction of the basin, only not switching for Madeira and Trombetas for LSTday_mean and Tapajós for LSTnight_mean.
These local effects could be linked to local differences in soil phosphorus distribution across the Amazon Basin. Yet, soil in western regions of the Amazon has been reported to contain significantly larger phosphorus concentrations than the northern, southern, eastern and central regions (Reichert et al., 2022), and this distribution does not align well with the sub-basins where we found a shift in effect. It could be that at the sub-basin scale, other factors supersede the effects of phosphorous limitation, such as water accessibility through, e.g., the roots (Chen et al., 2008; Fan et al., 2017; Gavrilescu, 2021).

## 310    4.3    Plant traits and regulation of soil moisture content

We found no effect of plant traits on soil moisture content, except for a few weak relationships for Solimões and Tapajós sub-basins. Water table depth exerts a strong control on water processes in the Amazon (O'connor et al., 2019). High canopy cover can reduce soil moisture content via increases in both evaporation and interception of precipitation (Zhang et al., 2020; Dai et al., 2022) and root uptake (Chen et al., 2008; Fan et al., 2017; Gavrilescu, 2021). Yet measurements of root traits are
few, making it difficult for our models to capture these effects using mostly gas exchange and photosynthesis related traits.

## 4.4    Plant trait effects at extreme values

Overall the quantile regression results show a very strong effect and potential feedback mediated by plant traits on water process parameters. We found that plant traits exert a stronger control at extreme low values of ET and at extreme high values of PET, VPD and LST. These results are consistent, as we know that VPD and LST decrease when there are water exchanges.
These results that with further increases in VPD or LST and further decreases in ET and PET due to ongoing climate changes plant trait distributions may have a yet underexplored capacity to exacerbate water fluxes. In Figure 6, we show the potentially



problematic areas in the Amazon basin for the time period we examined, i.e. the locations with extreme low values of ET and PET and extreme high values of values of VPD and LST. These values are not restricted to certain areas of the basin, yet in Xingu and Tapajós there is already some spatial overlap between low ET and PET and high LST and VPD where plant traits

may already have this double mediating effect.

Our analyses, however, are a function of certain choices that could have affected the results. We examined plant traits that describe phenology (SLA, LDMC) and biochemistry (LPC, LNC). Other plant traits also play significant roles in regulating evapotranspiration, such as stomatal conductance per leaf area or leaf dry mass (Wehr et al., 2017; Ding et al.) and root traits (fine root dry mass, rooting depth) (Fort et al., 2017; Delfin et al., 2021; Shao et al., 2022). Yet data for the later set of traits

is currently limited as evidenced by the few entries in the TRY database (Kattge et al., 2020), and predicting their distribution using remote sensing is not well established. We used model outputs of plant trait values generated using machine learning (Moreno-Martínez et al., 2018), which has a reasonable agreement with in-situ measurements. Yet, model performance and validation could be improved when more data from in situ sampling is acquired and local model performance metrics can be calculated. The data we used corresponds to averages over ten years, which may dillute the finer temporal dynamics between

plants and water cycle parameters. For example, the Amazon exhibits a strong seasonal variation in ET with annual minima between April-June and maxima between August-October and with peaks during the dry season and lower ET during the wet season (Baker et al., 2021), which may not be captured with averages over a decade. Yet, given that we already identify some trends even with this coarse average data, it suggests that finer temporal resolution data will likely produce even stronger relationships, and these could complement our understanding of the role of plant traits in mediating water processes. As

additional trait data becomes available, for example from new remote sensing products (Ustin and Middleton, 2021) and we better understand the role of biodiversity on ecosystem functioning (Yan et al., 2023), more clarification will emerge.

## 5   Conclusion

We found that plant traits have significant effects on the regulation of atmospheric water and the regulation of land surface temperature but not on the regulation of soil moisture content. The most important effects were driven by SLA, LDMC and

LPC, and these relationships tended to be stronger at the sub-basin level. We also found that plant traits could potentially have a double effect in mediating extremely low ET and extremely high PET, VPD and LST, which is not well understood and could accelerate the loss of resilience of the Amazon forest. Also, the stronger local effects are interesting and potentially resulting from localized impacts of land use change and deforestation, which might scale up to water fluxes potential to regulate climate at local and global scales. Together, these results suggest that plant traits can in fact be an important component of the regulation

of the processes that maintain the Amazon forest, potentially amplifying or mediating feedbacks that drive the response of the Amazon to ongoing and future global changes.

*Competing interests.*   The contact author has declared that none of the authors has any competing interests.





*Author contributions.* KN and MJS designed the study; KN performed the data analysis; KN and MJS wrote the manuscript.

*Code and data availability.* All raw data and code can be provided by the corresponding authors upon request.





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



## Appendix A:  Quantile Regression Results



**Table A1.** Quantile regression results (Q5) for the Amazon Basin

| PARAMETER | STAT | SLA | LDMC | LNC | LPC |
|---|---|---|---|---|---|
| ET_mean | $R^2$ | 0.175 | 0.117 | 0.002 | 0.002 |
| | $\beta^*$ | -0.808 | 0.838 | -0.074 | -0.101 |
| | t-value | -65.432 | 94.509 | -4.668 | -11.113 |
| | p-value | 0.000 | 0.000 | 0.000 | 0.000 |
| ET_sd | $R^2$ | 0.003 | 0.000 | 0.001 | 0.025 |
| | $\beta^*$ | 0.032 | 0.005 | 0.011 | -0.066 |
| | t-value | 32.633 | 3.331 | 9.141 | -48.902 |
| | p-value | 0.000 | 0.001 | 0.000 | 0.000 |
| PET_mean | $R^2$ | 0.001 | 0.004 | 0.008 | 0.059 |
| | $\beta^*$ | 0.032 | 0.052 | 0.073 | -0.240 |
| | t-value | 10.733 | 13.939 | 21.868 | -63.852 |
| | p-value | 0.000 | 0.000 | 0.000 | 0.000 |
| PET_sd | $R^2$ | 0.000 | 0.011 | 0.000 | 0.051 |
| | $\beta^*$ | -0.005 | 0.043 | 0.002 | -0.107 |
| | t-value | -4.706 | 33.890 | 0.861 | -73.764 |
| | p-value | 0.000 | 0.000 | 0.389 | 0.000 |
| SM_mean | $R^2$ | 0.020 | 0.045 | 0.000 | 0.039 |
| | $\beta^*$ | -0.209 | 0.265 | 0.008 | -0.208 |
| | t-value | -35.196 | 58.740 | 0.720 | -36.406 |
| | p-value | 0.000 | 0.000 | 0.472 | 0.000 |
| SM_sd | $R^2$ | 0.031 | 0.007 | 0.002 | 0.031 |
| | $\beta^*$ | 0.202 | -0.068 | 0.049 | -0.154 |
| | t-value | 84.185 | -14.741 | 26.626 | -50.242 |
| | p-value | 0.000 | 0.000 | 0.000 | 0.000 |
| VPD_mean | $R^2$ | 0.000 | 0.019 | 0.006 | 0.044 |
| | $\beta^*$ | 0.010 | 0.131 | 0.075 | -0.213 |
| | t-value | 4.197 | 47.270 | 20.805 | -63.692 |
| | p-value | 0.000 | 0.000 | 0.000 | 0.000 |
| VPD_sd | $R^2$ | 0.010 | 0.000 | 0.002 | 0.042 |
| | $\beta^*$ | 0.070 | -0.003 | 0.026 | -0.106 |
| | t-value | 92.698 | -2.497 | 32.507 | -94.517 |
| | p-value | 0.000 | 0.013 | 0.000 | 0.000 |
| LSTday_mean | $R^2$ | 0.004 | 0.013 | 0.065 | 0.071 |
| | $\beta^*$ | 0.219 | 0.258 | 0.729 | -0.606 |
| | t-value | 11.486 | 11.528 | 27.972 | -30.745 |
| | p-value | 0.000 | 0.000 | 0.000 | 0.000 |
| LSTday_sd | $R^2$ | 0.037 | 0.013 | 0.000 | 0.001 |
| | $\beta^*$ | 0.163 | -0.070 | -0.013 | -0.012 |
| | t-value | 180.588 | -37.518 | -15.786 | -11.824 |
| | p-value | 0.000 | 0.000 | 0.000 | 0.000 |
| LSTnight_mean | $R^2$ | 0.018 | 0.137 | 0.009 | 0.224 |
| | $\beta^*$ | -0.498 | 0.930 | 0.354 | -1.270 |
| | t-value | -11.195 | 35.901 | 7.064 | -109.380 |
| | p-value | 0.000 | 0.000 | 0.000 | 0.000 |
| LSTnight_sd | $R^2$ | 0.003 | 0.000 | 0.000 | 0.001 |
| | $\beta^*$ | 0.042 | -0.009 | -0.002 | 0.021 |
| | t-value | 27.660 | -6.224 | -1.638 | 12.831 |
| | p-value | 0.000 | 0.000 | 0.101 | 0.000 |





**Table A2.** Quantile regression results (Q5) for the Amazonas Sub-basin

| PARAMETER | STAT | SLA | LDMC | LNC | LPC |
|---|---|---|---|---|---|
| ET_mean | $R^2$ | 0.065 | 0.161 | 0.118 | 0.009 |
| | $\beta^*$ | -0.754 | 1.497 | 0.797 | -0.218 |
| | t-value | -7.511 | 38.855 | 20.413 | -1.583 |
| | p-value | 0.000 | 0.000 | 0.000 | 0.114 |
| ET_sd | $R^2$ | 0.004 | 0.000 | 0.001 | 0.001 |
| | $\beta^*$ | 0.062 | -0.013 | 0.070 | 0.037 |
| | t-value | 3.689 | -0.547 | 7.341 | 2.649 |
| | p-value | 0.000 | 0.585 | 0.000 | 0.008 |
| PET_mean | $R^2$ | 0.047 | 0.132 | 0.094 | 0.008 |
| | $\beta^*$ | -0.447 | 1.167 | 0.629 | -0.151 |
| | t-value | -5.668 | 33.345 | 22.768 | -1.597 |
| | p-value | 0.000 | 0.000 | 0.000 | 0.110 |
| PET_sd | $R^2$ | 0.000 | 0.000 | 0.001 | 0.000 |
| | $\beta^*$ | 0.032 | 0.014 | 0.052 | -0.016 |
| | t-value | 2.157 | 1.240 | 5.548 | -0.871 |
| | p-value | 0.031 | 0.215 | 0.000 | 0.384 |
| SM_mean | $R^2$ | 0.014 | 0.006 | 0.001 | 0.000 |
| | $\beta^*$ | 0.110 | -0.079 | 0.063 | -0.009 |
| | t-value | 7.348 | -2.640 | 6.880 | -0.635 |
| | p-value | 0.000 | 0.008 | 0.000 | 0.525 |
| SM_sd | $R^2$ | 0.003 | 0.002 | 0.000 | 0.000 |
| | $\beta^*$ | -0.050 | 0.084 | 0.036 | -0.013 |
| | t-value | -2.868 | 7.413 | 3.053 | -0.898 |
| | p-value | 0.004 | 0.000 | 0.002 | 0.369 |
| VPD_mean | $R^2$ | 0.001 | 0.001 | 0.003 | 0.014 |
| | $\beta^*$ | -0.032 | 0.041 | 0.122 | -0.165 |
| | t-value | -1.931 | 2.998 | 9.553 | -5.264 |
| | p-value | 0.053 | 0.003 | 0.000 | 0.000 |
| VPD_sd | $R^2$ | 0.009 | 0.010 | 0.015 | 0.005 |
| | $\beta^*$ | -0.109 | 0.143 | 0.222 | -0.084 |
| | t-value | -8.529 | 14.281 | 25.458 | -5.107 |
| | p-value | 0.000 | 0.000 | 0.000 | 0.000 |
| LSTday_mean | $R^2$ | 0.069 | 0.013 | 0.019 | 0.048 |
| | $\beta^*$ | 0.319 | -0.086 | 0.225 | 0.229 |
| | t-value | 37.188 | -4.042 | 55.027 | 30.859 |
| | p-value | 0.000 | 0.000 | 0.000 | 0.000 |
| LSTday_sd | $R^2$ | 0.012 | 0.001 | 0.002 | 0.001 |
| | $\beta^*$ | 0.101 | -0.028 | 0.062 | 0.026 |
| | t-value | 14.558 | -2.528 | 15.847 | 4.857 |
| | p-value | 0.000 | 0.012 | 0.000 | 0.000 |
| LSTnight_mean | $R^2$ | 0.055 | 0.024 | 0.004 | 0.018 |
| | $\beta^*$ | 0.281 | -0.166 | 0.085 | 0.130 |
| | t-value | 40.577 | -11.052 | 26.992 | 28.954 |
| | p-value | 0.000 | 0.000 | 0.000 | 0.000 |
| LSTnight_sd | $R^2$ | 0.000 | 0.000 | 0.000 | 0.003 |
| | $\beta^*$ | -0.015 | 0.001 | 0.000 | -0.038 |
| | t-value | -0.873 | 0.022 | 0.015 | -1.543 |
| | p-value | 0.383 | 0.983 | 0.988 | 0.123 |




**Table A3.** Quantile regression results (Q5) for the Madeira Sub-basin

| PARAMETER | STAT | SLA | LDMC | LNC | LPC |
|---|---|---|---|---|---|
| ET_mean | $R^2$ | 0.067 | 0.051 | 0.001 | 0.004 |
| | $\beta^*$ | -0.440 | 0.563 | -0.028 | -0.079 |
| | t-value | -35.560 | 88.908 | -2.366 | -10.082 |
| | p-value | 0.000 | 0.000 | 0.018 | 0.000 |
| ET_sd | $R^2$ | 0.006 | 0.001 | 0.010 | 0.003 |
| | $\beta^*$ | 0.049 | -0.010 | 0.072 | -0.024 |
| | t-value | 15.875 | -2.623 | 34.510 | -6.775 |
| | p-value | 0.000 | 0.009 | 0.000 | 0.000 |
| PET_mean | $R^2$ | 0.045 | 0.000 | 0.036 | 0.058 |
| | $\beta^*$ | 0.292 | -0.010 | 0.243 | -0.199 |
| | t-value | 40.134 | -0.728 | 38.751 | -23.197 |
| | p-value | 0.000 | 0.467 | 0.000 | 0.000 |
| PET_sd | $R^2$ | 0.019 | 0.020 | 0.000 | 0.021 |
| | $\beta^*$ | -0.106 | 0.116 | -0.003 | -0.101 |
| | t-value | -21.557 | 28.523 | -0.416 | -15.276 |
| | p-value | 0.000 | 0.000 | 0.677 | 0.000 |
| SM_mean | $R^2$ | 0.044 | 0.052 | 0.002 | 0.056 |
| | $\beta^*$ | -0.363 | 0.405 | 0.050 | -0.303 |
| | t-value | -21.750 | 30.356 | 1.769 | -13.815 |
| | p-value | 0.000 | 0.000 | 0.077 | 0.000 |
| SM_sd | $R^2$ | 0.006 | 0.001 | 0.007 | 0.008 |
| | $\beta^*$ | 0.099 | -0.032 | 0.119 | -0.085 |
| | t-value | 12.206 | -2.946 | 18.063 | -9.339 |
| | p-value | 0.000 | 0.003 | 0.000 | 0.000 |
| VPD_mean | $R^2$ | 0.010 | 0.000 | 0.075 | 0.012 |
| | $\beta^*$ | 0.079 | -0.016 | 0.377 | -0.098 |
| | t-value | 12.011 | -1.870 | 76.390 | -8.455 |
| | p-value | 0.000 | 0.061 | 0.000 | 0.000 |
| VPD_sd | $R^2$ | 0.084 | 0.009 | 0.023 | 0.056 |
| | $\beta^*$ | 0.445 | -0.100 | 0.190 | -0.179 |
| | t-value | 80.450 | -8.630 | 35.768 | -20.471 |
| | p-value | 0.000 | 0.000 | 0.000 | 0.000 |
| LSTday_mean | $R^2$ | 0.078 | 0.011 | 0.196 | 0.058 |
| | $\beta^*$ | 0.724 | -0.147 | 1.016 | -0.447 |
| | t-value | 25.730 | -2.044 | 46.608 | -7.573 |
| | p-value | 0.000 | 0.041 | 0.000 | 0.000 |
| LSTday_sd | $R^2$ | 0.147 | 0.034 | 0.002 | 0.007 |
| | $\beta^*$ | 0.344 | -0.152 | 0.065 | 0.066 |
| | t-value | 98.267 | -19.609 | 41.444 | 20.594 |
| | p-value | 0.000 | 0.000 | 0.000 | 0.000 |
| LSTnight_mean | $R^2$ | 0.002 | 0.011 | 0.098 | 0.185 |
| | $\beta^*$ | 0.129 | 0.282 | 1.028 | -1.121 |
| | t-value | 3.423 | 5.970 | 20.116 | -20.539 |
| | p-value | 0.001 | 0.000 | 0.000 | 0.000 |
| LSTnight_sd | $R^2$ | 0.010 | 0.001 | 0.003 | 0.002 |
| | $\beta^*$ | 0.064 | -0.023 | 0.071 | 0.034 |
| | t-value | 16.209 | -5.731 | 39.879 | 10.034 |
| | p-value | 0.000 | 0.000 | 0.000 | 0.000 |




**Table A4.** Quantile regression results (Q5) for the Negro Sub-basin

| PARAMETER | STAT | SLA | LDMC | LNC | LPC |
|---|---|---|---|---|---|
| ET_mean | $R^2$ | 0.303 | 0.301 | 0.004 | 0.006 |
| | $\beta^*$ | -1.068 | 0.950 | -0.101 | -0.214 |
| | t-value | -21.614 | 15.851 | -1.070 | -3.375 |
| | p-value | 0.000 | 0.000 | 0.285 | 0.001 |
| ET_sd | $R^2$ | 0.002 | 0.005 | 0.004 | 0.000 |
| | $\beta^*$ | 0.025 | -0.038 | 0.033 | -0.006 |
| | t-value | 6.869 | -5.811 | 9.515 | -1.449 |
| | p-value | 0.000 | 0.000 | 0.000 | 0.147 |
| PET_mean | $R^2$ | 0.010 | 0.018 | 0.061 | 0.002 |
| | $\beta^*$ | 0.146 | -0.160 | 0.284 | 0.065 |
| | t-value | 12.581 | -6.071 | 27.318 | 5.255 |
| | p-value | 0.000 | 0.000 | 0.000 | 0.000 |
| PET_sd | $R^2$ | 0.003 | 0.003 | 0.000 | 0.008 |
| | $\beta^*$ | 0.027 | -0.023 | 0.004 | -0.037 |
| | t-value | 8.002 | -4.313 | 1.322 | -9.239 |
| | p-value | 0.000 | 0.000 | 0.186 | 0.000 |
| SM_mean | $R^2$ | 0.002 | 0.001 | 0.004 | 0.012 |
| | $\beta^*$ | 0.061 | -0.054 | 0.103 | -0.151 |
| | t-value | 4.089 | -4.612 | 8.562 | -8.731 |
| | p-value | 0.000 | 0.000 | 0.000 | 0.000 |
| SM_sd | $R^2$ | 0.009 | 0.003 | 0.000 | 0.001 |
| | $\beta^*$ | 0.063 | -0.036 | 0.011 | -0.015 |
| | t-value | 14.538 | -6.000 | 3.507 | -3.484 |
| | p-value | 0.000 | 0.000 | 0.000 | 0.000 |
| VPD_mean | $R^2$ | 0.086 | 0.002 | 0.045 | 0.023 |
| | $\beta^*$ | 0.350 | -0.036 | 0.129 | 0.088 |
| | t-value | 132.094 | -8.633 | 48.082 | 21.385 |
| | p-value | 0.000 | 0.000 | 0.000 | 0.000 |
| VPD_sd | $R^2$ | 0.041 | 0.019 | 0.000 | 0.059 |
| | $\beta^*$ | 0.275 | -0.141 | 0.007 | -0.134 |
| | t-value | 164.138 | -31.229 | 4.045 | -51.983 |
| | p-value | 0.000 | 0.000 | 0.000 | 0.000 |
| LSTday_mean | $R^2$ | 0.001 | 0.007 | 0.001 | 0.051 |
| | $\beta^*$ | -0.048 | 0.174 | 0.102 | -0.289 |
| | t-value | -3.662 | 17.510 | 9.258 | -15.165 |
| | p-value | 0.000 | 0.000 | 0.000 | 0.000 |
| LSTday_sd | $R^2$ | 0.002 | 0.013 | 0.009 | 0.038 |
| | $\beta^*$ | 0.040 | -0.076 | -0.054 | -0.091 |
| | t-value | 23.528 | -15.450 | -13.252 | -27.816 |
| | p-value | 0.000 | 0.000 | 0.000 | 0.000 |
| LSTnight_mean | $R^2$ | 0.000 | 0.016 | 0.006 | 0.085 |
| | $\beta^*$ | -0.052 | 0.662 | 0.272 | -0.569 |
| | t-value | -2.823 | 39.116 | 18.689 | -26.405 |
| | p-value | 0.005 | 0.000 | 0.000 | 0.000 |
| LSTnight_sd | $R^2$ | 0.029 | 0.029 | 0.016 | 0.014 |
| | $\beta^*$ | -0.204 | 0.331 | -0.128 | 0.133 |
| | t-value | -19.501 | 67.896 | -6.452 | 14.328 |
| | p-value | 0.000 | 0.000 | 0.000 | 0.000 |



**Table A5.** Quantile regression results (Q5) for the Solimoes Sub-basin

| PARAMETER | STAT | SLA | LDMC | LNC | LPC |
|---|---|---|---|---|---|
| ET_mean | $R^2$ | 0.087 | 0.177 | 0.000 | 0.107 |
| | $\beta^*$ | -0.480 | 0.703 | -0.009 | -0.815 |
| | t-value | -13.457 | 30.799 | -0.250 | -25.967 |
| | p-value | 0.000 | 0.000 | 0.803 | 0.000 |
| ET_sd | $R^2$ | 0.010 | 0.016 | 0.000 | 0.033 |
| | $\beta^*$ | -0.037 | 0.059 | -0.005 | -0.075 |
| | t-value | -13.056 | 41.512 | -1.454 | -31.051 |
| | p-value | 0.000 | 0.000 | 0.146 | 0.000 |
| PET_mean | $R^2$ | 0.012 | 0.032 | 0.002 | 0.067 |
| | $\beta^*$ | -0.102 | 0.188 | 0.041 | -0.363 |
| | t-value | -9.297 | 29.387 | 3.305 | -47.555 |
| | p-value | 0.000 | 0.000 | 0.001 | 0.000 |
| PET_sd | $R^2$ | 0.038 | 0.064 | 0.002 | 0.076 |
| | $\beta^*$ | -0.067 | 0.084 | -0.016 | -0.124 |
| | t-value | -24.776 | 57.334 | -3.447 | -52.924 |
| | p-value | 0.000 | 0.000 | 0.001 | 0.000 |
| SM_mean | $R^2$ | 0.041 | 0.067 | 0.000 | 0.049 |
| | $\beta^*$ | -0.201 | 0.286 | -0.017 | -0.263 |
| | t-value | -15.279 | 36.006 | -1.027 | -27.065 |
| | p-value | 0.000 | 0.000 | 0.304 | 0.000 |
| SM_sd | $R^2$ | 0.008 | 0.004 | 0.000 | 0.010 |
| | $\beta^*$ | 0.093 | -0.051 | 0.018 | -0.087 |
| | t-value | 32.782 | -8.956 | 6.710 | -23.874 |
| | p-value | 0.000 | 0.000 | 0.000 | 0.000 |
| VPD_mean | $R^2$ | 0.009 | 0.043 | 0.016 | 0.090 |
| | $\beta^*$ | -0.174 | 0.522 | 0.255 | -0.531 |
| | t-value | -4.809 | 21.521 | 10.912 | -42.037 |
| | p-value | 0.000 | 0.000 | 0.000 | 0.000 |
| VPD_sd | $R^2$ | 0.001 | 0.006 | 0.007 | 0.019 |
| | $\beta^*$ | -0.031 | 0.066 | 0.097 | -0.151 |
| | t-value | -6.000 | 14.462 | 23.230 | -26.083 |
| | p-value | 0.000 | 0.000 | 0.000 | 0.000 |
| LSTday_mean | $R^2$ | 0.001 | 0.001 | 0.055 | 0.018 |
| | $\beta^*$ | 0.078 | 0.065 | 0.494 | -0.352 |
| | t-value | 3.162 | 2.154 | 20.298 | -10.654 |
| | p-value | 0.002 | 0.031 | 0.000 | 0.000 |
| LSTday_sd | $R^2$ | 0.037 | 0.016 | 0.001 | 0.003 |
| | $\beta^*$ | 0.123 | -0.076 | 0.022 | -0.023 |
| | t-value | 94.064 | -25.212 | 29.061 | -18.361 |
| | p-value | 0.000 | 0.000 | 0.000 | 0.000 |
| LSTnight_mean | $R^2$ | 0.080 | 0.145 | 0.000 | 0.157 |
| | $\beta^*$ | -0.375 | 0.452 | -0.014 | -0.788 |
| | t-value | -16.443 | 26.220 | -0.560 | -41.466 |
| | p-value | 0.000 | 0.000 | 0.575 | 0.000 |
| LSTnight_sd | $R^2$ | 0.000 | 0.000 | 0.006 | 0.001 |
| | $\beta^*$ | 0.012 | 0.004 | 0.049 | 0.014 |
| | t-value | 6.835 | 2.873 | 30.481 | 7.249 |
| | p-value | 0.000 | 0.004 | 0.000 | 0.000 |





**Table A6.** Quantile regression results (Q5) for the Tapajos Sub-basin

| PARAMETER | STAT | SLA | LDMC | LNC | LPC |
|---|---|---|---|---|---|
| ET_mean | $R^2$ | 0.108 | 0.174 | 0.004 | 0.157 |
| | $\beta^*$ | -0.752 | 0.842 | -0.071 | -0.744 |
| | t-value | -51.136 | 58.497 | -2.987 | -31.736 |
| | p-value | 0.000 | 0.000 | 0.003 | 0.000 |
| ET_sd | $R^2$ | 0.040 | 0.032 | 0.000 | 0.037 |
| | $\beta^*$ | 0.176 | -0.161 | 0.006 | 0.147 |
| | t-value | 28.832 | -13.878 | 0.802 | 26.934 |
| | p-value | 0.000 | 0.000 | 0.422 | 0.000 |
| PET_mean | $R^2$ | 0.061 | 0.027 | 0.000 | 0.026 |
| | $\beta^*$ | 0.403 | -0.328 | 0.002 | 0.241 |
| | t-value | 40.456 | -12.934 | 0.100 | 23.497 |
| | p-value | 0.000 | 0.000 | 0.920 | 0.000 |
| PET_sd | $R^2$ | 0.005 | 0.002 | 0.001 | 0.000 |
| | $\beta^*$ | 0.110 | -0.039 | 0.030 | -0.011 |
| | t-value | 12.426 | -2.577 | 3.538 | -1.123 |
| | p-value | 0.000 | 0.010 | 0.000 | 0.262 |
| SM_mean | $R^2$ | 0.010 | 0.006 | 0.000 | 0.010 |
| | $\beta^*$ | 0.144 | -0.079 | -0.015 | 0.127 |
| | t-value | 11.441 | -3.638 | -0.994 | 10.165 |
| | p-value | 0.000 | 0.000 | 0.320 | 0.000 |
| SM_sd | $R^2$ | 0.013 | 0.008 | 0.002 | 0.007 |
| | $\beta^*$ | 0.142 | -0.085 | 0.057 | 0.107 |
| | t-value | 10.290 | -3.268 | 3.568 | 7.806 |
| | p-value | 0.000 | 0.001 | 0.000 | 0.000 |
| VPD_mean | $R^2$ | 0.037 | 0.014 | 0.001 | 0.001 |
| | $\beta^*$ | 0.254 | -0.185 | -0.038 | 0.046 |
| | t-value | 26.769 | -8.887 | -1.640 | 6.875 |
| | p-value | 0.000 | 0.000 | 0.101 | 0.000 |
| VPD_sd | $R^2$ | 0.034 | 0.020 | 0.001 | 0.006 |
| | $\beta^*$ | 0.166 | -0.172 | -0.018 | 0.110 |
| | t-value | 27.789 | -11.818 | -1.058 | 27.445 |
| | p-value | 0.000 | 0.000 | 0.290 | 0.000 |
| LSTday_mean | $R^2$ | 0.107 | 0.061 | 0.003 | 0.067 |
| | $\beta^*$ | 0.300 | -0.259 | 0.027 | 0.251 |
| | t-value | 73.182 | -33.092 | 7.808 | 81.297 |
| | p-value | 0.000 | 0.000 | 0.000 | 0.000 |
| LSTday_sd | $R^2$ | 0.106 | 0.085 | 0.000 | 0.098 |
| | $\beta^*$ | 0.263 | -0.233 | -0.002 | 0.230 |
| | t-value | 66.013 | -33.234 | -0.397 | 77.713 |
| | p-value | 0.000 | 0.000 | 0.692 | 0.000 |
| LSTnight_mean | $R^2$ | 0.016 | 0.031 | 0.001 | 0.099 |
| | $\beta^*$ | -0.201 | 0.313 | -0.043 | -0.481 |
| | t-value | -18.865 | 30.954 | -2.908 | -29.750 |
| | p-value | 0.000 | 0.000 | 0.004 | 0.000 |
| LSTnight_sd | $R^2$ | 0.017 | 0.011 | 0.001 | 0.007 |
| | $\beta^*$ | 0.138 | -0.111 | -0.020 | 0.096 |
| | t-value | 15.109 | -7.867 | -1.369 | 11.631 |
| | p-value | 0.000 | 0.000 | 0.171 | 0.000 |





**Table A7.** Quantile regression results (Q5) for the Trombetas Sub-basin

| PARAMETER | STAT | SLA | LDMC | LNC | LPC |
|---|---|---|---|---|---|
| ET_mean | $R^2$ | 0.017 | 0.027 | 0.094 | 0.008 |
| | $\beta^*$ | -0.215 | 0.277 | 0.820 | -0.209 |
| | t-value | -1.394 | 2.067 | 23.355 | -1.574 |
| | p-value | 0.163 | 0.039 | 0.000 | 0.116 |
| ET_sd | $R^2$ | 0.002 | 0.004 | 0.000 | 0.002 |
| | $\beta^*$ | 0.065 | -0.077 | 0.030 | -0.039 |
| | t-value | 5.305 | -3.628 | 2.635 | -2.837 |
| | p-value | 0.000 | 0.000 | 0.008 | 0.005 |
| PET_mean | $R^2$ | 0.001 | 0.000 | 0.010 | 0.032 |
| | $\beta^*$ | -0.022 | 0.005 | 0.170 | -0.207 |
| | t-value | -1.526 | 0.336 | 18.237 | -12.823 |
| | p-value | 0.127 | 0.737 | 0.000 | 0.000 |
| PET_sd | $R^2$ | 0.000 | 0.002 | 0.000 | 0.012 |
| | $\beta^*$ | -0.005 | -0.043 | -0.021 | -0.106 |
| | t-value | -0.694 | -3.475 | -1.964 | -13.457 |
| | p-value | 0.488 | 0.001 | 0.050 | 0.000 |
| SM_mean | $R^2$ | 0.002 | 0.000 | 0.001 | 0.017 |
| | $\beta^*$ | 0.052 | 0.014 | -0.034 | 0.137 |
| | t-value | 3.693 | 1.031 | -1.958 | 11.177 |
| | p-value | 0.000 | 0.303 | 0.050 | 0.000 |
| SM_sd | $R^2$ | 0.000 | 0.000 | 0.006 | 0.007 |
| | $\beta^*$ | 0.004 | 0.007 | 0.076 | -0.083 |
| | t-value | 0.384 | 0.526 | 11.651 | -7.649 |
| | p-value | 0.701 | 0.599 | 0.000 | 0.000 |
| VPD_mean | $R^2$ | 0.000 | 0.001 | 0.000 | 0.066 |
| | $\beta^*$ | 0.032 | -0.055 | 0.020 | -0.275 |
| | t-value | 1.651 | -1.895 | 0.822 | -8.562 |
| | p-value | 0.099 | 0.058 | 0.411 | 0.000 |
| VPD_sd | $R^2$ | 0.002 | 0.015 | 0.003 | 0.223 |
| | $\beta^*$ | 0.129 | -0.133 | 0.092 | -0.626 |
| | t-value | 9.700 | -2.388 | 5.366 | -32.900 |
| | p-value | 0.000 | 0.017 | 0.000 | 0.000 |
| LSTday_mean | $R^2$ | 0.022 | 0.051 | 0.002 | 0.067 |
| | $\beta^*$ | 0.308 | 0.560 | -0.078 | 0.461 |
| | t-value | 20.908 | 42.331 | -3.299 | 36.731 |
| | p-value | 0.000 | 0.000 | 0.001 | 0.000 |
| LSTday_sd | $R^2$ | 0.012 | 0.015 | 0.001 | 0.026 |
| | $\beta^*$ | 0.132 | -0.086 | -0.031 | 0.095 |
| | t-value | 29.088 | -6.106 | -4.985 | 14.940 |
| | p-value | 0.000 | 0.000 | 0.000 | 0.000 |
| LSTnight_mean | $R^2$ | 0.010 | 0.126 | 0.018 | 0.068 |
| | $\beta^*$ | -0.133 | 0.915 | -0.191 | 0.427 |
| | t-value | -8.228 | 81.473 | -4.803 | 42.191 |
| | p-value | 0.000 | 0.000 | 0.000 | 0.000 |
| LSTnight_sd | $R^2$ | 0.002 | 0.005 | 0.007 | 0.010 |
| | $\beta^*$ | -0.067 | 0.183 | -0.078 | 0.145 |
| | t-value | -5.615 | 23.676 | -2.986 | 14.385 |
| | p-value | 0.000 | 0.000 | 0.003 | 0.000 |



**Table A8.** Quantile regression results (Q5) for the Xingu Sub-basin

| PARAMETER | STAT | SLA | LDMC | LNC | LPC |
|---|---|---|---|---|---|
| ET_mean | $R^2$ | 0.179 | 0.197 | 0.004 | 0.148 |
| | $\beta^*$ | -0.748 | 0.941 | -0.101 | -0.646 |
| | t-value | -29.780 | 46.573 | -4.657 | -17.893 |
| | p-value | 0.000 | 0.000 | 0.000 | 0.000 |
| ET_sd | $R^2$ | 0.003 | 0.000 | 0.003 | 0.029 |
| | $\beta^*$ | 0.043 | 0.004 | 0.050 | 0.152 |
| | t-value | 10.511 | 0.531 | 12.828 | 32.610 |
| | p-value | 0.000 | 0.595 | 0.000 | 0.000 |
| PET_mean | $R^2$ | 0.031 | 0.048 | 0.012 | 0.001 |
| | $\beta^*$ | -0.323 | 0.468 | 0.186 | 0.024 |
| | t-value | -30.229 | 63.124 | 14.592 | 1.454 |
| | p-value | 0.000 | 0.000 | 0.000 | 0.146 |
| PET_sd | $R^2$ | 0.001 | 0.004 | 0.003 | 0.005 |
| | $\beta^*$ | -0.022 | 0.050 | 0.048 | 0.053 |
| | t-value | -4.611 | 12.918 | 12.792 | 10.980 |
| | p-value | 0.000 | 0.000 | 0.000 | 0.000 |
| SM_mean | $R^2$ | 0.000 | 0.001 | 0.001 | 0.000 |
| | $\beta^*$ | -0.005 | 0.027 | 0.031 | -0.021 |
| | t-value | -0.379 | 2.054 | 2.990 | -1.313 |
| | p-value | 0.705 | 0.040 | 0.003 | 0.189 |
| SM_sd | $R^2$ | 0.007 | 0.002 | 0.002 | 0.009 |
| | $\beta^*$ | 0.096 | -0.043 | 0.054 | 0.092 |
| | t-value | 9.906 | -2.615 | 6.743 | 9.536 |
| | p-value | 0.000 | 0.009 | 0.000 | 0.000 |
| VPD_mean | $R^2$ | 0.002 | 0.000 | 0.002 | 0.022 |
| | $\beta^*$ | 0.041 | -0.007 | 0.053 | 0.110 |
| | t-value | 15.442 | -1.084 | 26.551 | 33.376 |
| | p-value | 0.000 | 0.278 | 0.000 | 0.000 |
| VPD_sd | $R^2$ | 0.005 | 0.008 | 0.007 | 0.001 |
| | $\beta^*$ | -0.058 | 0.075 | 0.090 | 0.022 |
| | t-value | -12.917 | 19.999 | 21.852 | 3.523 |
| | p-value | 0.000 | 0.000 | 0.000 | 0.000 |
| LSTday_mean | $R^2$ | 0.008 | 0.001 | 0.022 | 0.082 |
| | $\beta^*$ | 0.092 | 0.018 | 0.129 | 0.297 |
| | t-value | 23.402 | 3.735 | 51.558 | 83.990 |
| | p-value | 0.000 | 0.000 | 0.000 | 0.000 |
| LSTday_sd | $R^2$ | 0.060 | 0.027 | 0.002 | 0.008 |
| | $\beta^*$ | 0.213 | -0.105 | -0.020 | 0.088 |
| | t-value | 63.854 | -12.685 | -3.763 | 47.836 |
| | p-value | 0.000 | 0.000 | 0.000 | 0.000 |
| LSTnight_mean | $R^2$ | 0.052 | 0.067 | 0.001 | 0.072 |
| | $\beta^*$ | -0.359 | 0.430 | 0.084 | -0.387 |
| | t-value | -23.841 | 29.100 | 3.985 | -13.342 |
| | p-value | 0.000 | 0.000 | 0.000 | 0.000 |
| LSTnight_sd | $R^2$ | 0.009 | 0.004 | 0.000 | 0.027 |
| | $\beta^*$ | 0.110 | -0.070 | 0.003 | 0.152 |
| | t-value | 14.281 | -5.082 | 0.197 | 18.364 |
| | p-value | 0.000 | 0.000 | 0.844 | 0.000 |





**Table A9.** Quantile regression results (Q50) for the Amazon Basin

| PARAMETER | STAT | SLA | LDMC | LNC | LPC |
|---|---|---|---|---|---|
| ET_mean | $R^2$ | 0.066 | 0.052 | 0.003 | 0.003 |
| | $\beta^*$ | -0.331 | 0.349 | 0.055 | -0.045 |
| | t-value | -321.375 | 342.584 | 54.249 | -44.201 |
| | p-value | 0.000 | 0.000 | 0.000 | 0.000 |
| ET_sd | $R^2$ | 0.048 | 0.007 | 0.002 | 0.021 |
| | $\beta^*$ | 0.274 | -0.124 | 0.051 | -0.149 |
| | t-value | 156.400 | -66.321 | 26.734 | -79.343 |
| | p-value | 0.000 | 0.000 | 0.000 | 0.000 |
| PET_mean | $R^2$ | 0.095 | 0.020 | 0.012 | 0.075 |
| | $\beta^*$ | 0.478 | -0.346 | 0.218 | -0.384 |
| | t-value | 175.585 | -104.742 | 63.448 | -143.917 |
| | p-value | 0.000 | 0.000 | 0.000 | 0.000 |
| PET_sd | $R^2$ | 0.004 | 0.000 | 0.000 | 0.057 |
| | $\beta^*$ | 0.081 | 0.030 | 0.024 | -0.280 |
| | t-value | 27.662 | 10.143 | 8.044 | -125.249 |
| | p-value | 0.000 | 0.000 | 0.000 | 0.000 |
| SM_mean | $R^2$ | 0.017 | 0.024 | 0.001 | 0.015 |
| | $\beta^*$ | -0.173 | 0.268 | -0.064 | 0.206 |
| | t-value | -58.476 | 94.327 | -21.062 | 69.800 |
| | p-value | 0.000 | 0.000 | 0.000 | 0.000 |
| SM_sd | $R^2$ | 0.022 | 0.007 | 0.003 | 0.041 |
| | $\beta^*$ | 0.181 | -0.136 | 0.077 | -0.267 |
| | t-value | 68.222 | -49.132 | 26.803 | -104.884 |
| | p-value | 0.000 | 0.000 | 0.000 | 0.000 |
| VPD_mean | $R^2$ | 0.055 | 0.008 | 0.001 | 0.054 |
| | $\beta^*$ | 0.370 | -0.204 | 0.054 | -0.345 |
| | t-value | 104.734 | -46.443 | 12.730 | -122.196 |
| | p-value | 0.000 | 0.000 | 0.000 | 0.000 |
| VPD_sd | $R^2$ | 0.121 | 0.043 | 0.006 | 0.122 |
| | $\beta^*$ | 0.497 | -0.426 | 0.144 | -0.434 |
| | t-value | 189.782 | -140.419 | 39.422 | -230.688 |
| | p-value | 0.000 | 0.000 | 0.000 | 0.000 |
| LSTday_mean | $R^2$ | 0.045 | 0.009 | 0.001 | 0.000 |
| | $\beta^*$ | 0.224 | -0.124 | 0.021 | -0.000 |
| | t-value | 252.106 | -176.427 | 32.576 | -0.076 |
| | p-value | 0.000 | 0.000 | 0.000 | 0.939 |
| LSTday_sd | $R^2$ | 0.166 | 0.142 | 0.000 | 0.000 |
| | $\beta^*$ | 0.451 | -0.582 | 0.005 | -0.007 |
| | t-value | 397.440 | -498.066 | 3.670 | -5.709 |
| | p-value | 0.000 | 0.000 | 0.000 | 0.000 |
| LSTnight_mean | $R^2$ | 0.031 | 0.038 | 0.005 | 0.000 |
| | $\beta^*$ | -0.083 | 0.150 | -0.047 | -0.005 |
| | t-value | -161.390 | 280.706 | -84.502 | -7.852 |
| | p-value | 0.000 | 0.000 | 0.000 | 0.000 |
| LSTnight_sd | $R^2$ | 0.018 | 0.007 | 0.000 | 0.003 |
| | $\beta^*$ | 0.149 | -0.096 | -0.020 | 0.049 |
| | t-value | 84.119 | -54.122 | -10.949 | 27.755 |
| | p-value | 0.000 | 0.000 | 0.000 | 0.000 |





**Table A10.** Quantile regression results (Q50) for the Amazonas Sub-basin

| PARAMETER | STAT | SLA | LDMC | LNC | LPC |
|---|---|---|---|---|---|
| ET_mean | $R^2$ | 0.121 | 0.149 | 0.011 | 0.016 |
| | $\beta^*$ | -0.407 | 0.693 | 0.194 | -0.204 |
| | t-value | -49.899 | 86.330 | 17.455 | -21.093 |
| | p-value | 0.000 | 0.000 | 0.000 | 0.000 |
| ET_sd | $R^2$ | 0.021 | 0.006 | 0.000 | 0.002 |
| | $\beta^*$ | 0.189 | -0.134 | 0.004 | 0.047 |
| | t-value | 14.639 | -10.103 | 0.310 | 3.472 |
| | p-value | 0.000 | 0.000 | 0.756 | 0.001 |
| PET_mean | $R^2$ | 0.121 | 0.161 | 0.011 | 0.029 |
| | $\beta^*$ | -0.466 | 0.858 | 0.176 | -0.289 |
| | t-value | -47.944 | 91.650 | 13.569 | -27.982 |
| | p-value | 0.000 | 0.000 | 0.000 | 0.000 |
| PET_sd | $R^2$ | 0.015 | 0.003 | 0.000 | 0.004 |
| | $\beta^*$ | 0.159 | -0.068 | 0.030 | 0.086 |
| | t-value | 11.436 | -4.868 | 2.143 | 6.207 |
| | p-value | 0.000 | 0.000 | 0.032 | 0.000 |
| SM_mean | $R^2$ | 0.010 | 0.017 | 0.011 | 0.001 |
| | $\beta^*$ | 0.131 | -0.229 | -0.180 | -0.032 |
| | t-value | 9.498 | -16.539 | -13.080 | -2.320 |
| | p-value | 0.000 | 0.000 | 0.000 | 0.020 |
| SM_sd | $R^2$ | 0.001 | 0.000 | 0.000 | 0.000 |
| | $\beta^*$ | -0.058 | 0.015 | 0.015 | 0.008 |
| | t-value | -4.240 | 1.113 | 1.077 | 0.581 |
| | p-value | 0.000 | 0.266 | 0.282 | 0.561 |
| VPD_mean | $R^2$ | 0.005 | 0.001 | 0.001 | 0.003 |
| | $\beta^*$ | -0.105 | 0.062 | -0.037 | 0.095 |
| | t-value | -7.013 | 4.058 | -2.412 | 6.238 |
| | p-value | 0.000 | 0.000 | 0.016 | 0.000 |
| VPD_sd | $R^2$ | 0.019 | 0.003 | 0.013 | 0.071 |
| | $\beta^*$ | 0.191 | -0.071 | 0.301 | 0.419 |
| | t-value | 13.383 | -5.093 | 23.107 | 30.776 |
| | p-value | 0.000 | 0.000 | 0.000 | 0.000 |
| LSTday_mean | $R^2$ | 0.243 | 0.078 | 0.000 | 0.102 |
| | $\beta^*$ | 0.690 | -0.650 | -0.008 | 0.515 |
| | t-value | 71.863 | -43.725 | -0.469 | 44.435 |
| | p-value | 0.000 | 0.000 | 0.639 | 0.000 |
| LSTday_sd | $R^2$ | 0.127 | 0.035 | 0.001 | 0.039 |
| | $\beta^*$ | 0.463 | -0.373 | 0.058 | 0.206 |
| | t-value | 58.140 | -41.975 | 6.674 | 26.073 |
| | p-value | 0.000 | 0.000 | 0.000 | 0.000 |
| LSTnight_mean | $R^2$ | 0.187 | 0.151 | 0.005 | 0.063 |
| | $\beta^*$ | 0.533 | -0.745 | -0.160 | 0.397 |
| | t-value | 51.792 | -58.603 | -8.275 | 27.240 |
| | p-value | 0.000 | 0.000 | 0.000 | 0.000 |
| LSTnight_sd | $R^2$ | 0.008 | 0.000 | 0.016 | 0.009 |
| | $\beta^*$ | -0.112 | 0.022 | -0.279 | -0.133 |
| | t-value | -8.062 | 1.539 | -20.458 | -9.723 |
| | p-value | 0.000 | 0.124 | 0.000 | 0.000 |



**Table A11.** Quantile regression results (Q50) for the Madeira Sub-basin

| PARAMETER | STAT | SLA | LDMC | LNC | LPC |
|---|---|---|---|---|---|
| ET_mean | $R^2$ | 0.141 | 0.062 | 0.001 | 0.023 |
| | $\beta^*$ | -0.562 | 0.406 | 0.034 | -0.224 |
| | t-value | -201.277 | 131.827 | 9.595 | -57.495 |
| | p-value | 0.000 | 0.000 | 0.000 | 0.000 |
| ET_sd | $R^2$ | 0.064 | 0.019 | 0.009 | 0.001 |
| | $\beta^*$ | 0.343 | -0.237 | 0.170 | -0.046 |
| | t-value | 70.010 | -41.535 | 28.216 | -7.320 |
| | p-value | 0.000 | 0.000 | 0.000 | 0.000 |
| PET_mean | $R^2$ | 0.084 | 0.014 | 0.015 | 0.000 |
| | $\beta^*$ | 0.452 | -0.219 | 0.228 | -0.033 |
| | t-value | 81.643 | -38.639 | 43.598 | -6.404 |
| | p-value | 0.000 | 0.000 | 0.000 | 0.000 |
| PET_sd | $R^2$ | 0.006 | 0.000 | 0.000 | 0.002 |
| | $\beta^*$ | -0.095 | 0.028 | -0.031 | -0.058 |
| | t-value | -21.726 | 6.495 | -7.313 | -13.370 |
| | p-value | 0.000 | 0.000 | 0.000 | 0.000 |
| SM_mean | $R^2$ | 0.008 | 0.008 | 0.001 | 0.022 |
| | $\beta^*$ | -0.118 | 0.148 | 0.032 | -0.223 |
| | t-value | -20.405 | 25.576 | 5.489 | -39.189 |
| | p-value | 0.000 | 0.000 | 0.000 | 0.000 |
| SM_sd | $R^2$ | 0.006 | 0.002 | 0.005 | 0.003 |
| | $\beta^*$ | 0.098 | -0.070 | 0.114 | -0.071 |
| | t-value | 19.655 | -13.726 | 22.128 | -14.044 |
| | p-value | 0.000 | 0.000 | 0.000 | 0.000 |
| VPD_mean | $R^2$ | 0.076 | 0.008 | 0.003 | 0.002 |
| | $\beta^*$ | 0.415 | -0.149 | 0.087 | -0.065 |
| | t-value | 97.010 | -28.240 | 15.110 | -11.813 |
| | p-value | 0.000 | 0.000 | 0.000 | 0.000 |
| VPD_sd | $R^2$ | 0.093 | 0.025 | 0.036 | 0.000 |
| | $\beta^*$ | 0.444 | -0.371 | 0.447 | -0.024 |
| | t-value | 62.829 | -44.003 | 62.848 | -2.724 |
| | p-value | 0.000 | 0.000 | 0.000 | 0.006 |
| LSTday_mean | $R^2$ | 0.104 | 0.074 | 0.009 | 0.000 |
| | $\beta^*$ | 0.305 | -0.368 | 0.126 | 0.008 |
| | t-value | 129.214 | -153.791 | 58.974 | 4.184 |
| | p-value | 0.000 | 0.000 | 0.000 | 0.000 |
| LSTday_sd | $R^2$ | 0.235 | 0.138 | 0.000 | 0.072 |
| | $\beta^*$ | 0.652 | -0.645 | 0.015 | 0.374 |
| | t-value | 187.101 | -148.365 | 2.345 | 86.887 |
| | p-value | 0.000 | 0.000 | 0.019 | 0.000 |
| LSTnight_mean | $R^2$ | 0.016 | 0.008 | 0.000 | 0.043 |
| | $\beta^*$ | -0.074 | 0.083 | -0.005 | -0.162 |
| | t-value | -42.842 | 45.490 | -2.753 | -97.785 |
| | p-value | 0.000 | 0.000 | 0.006 | 0.000 |
| LSTnight_sd | $R^2$ | 0.069 | 0.012 | 0.001 | 0.009 |
| | $\beta^*$ | 0.383 | -0.191 | -0.038 | 0.129 |
| | t-value | 84.109 | -35.139 | -6.372 | 22.657 |
| | p-value | 0.000 | 0.000 | 0.000 | 0.000 |



**Table A12.** Quantile regression results (Q50) for the Negro Sub-basin

| PARAMETER | STAT | SLA | LDMC | LNC | LPC |
|---|---|---|---|---|---|
| ET_mean | $R^2$ | 0.010 | 0.023 | 0.008 | 0.000 |
| | $\beta^*$ | -0.132 | 0.237 | 0.081 | 0.002 |
| | t-value | -34.745 | 63.096 | 23.623 | 0.607 |
| | p-value | 0.000 | 0.000 | 0.000 | 0.544 |
| ET_sd | $R^2$ | 0.027 | 0.048 | 0.001 | 0.000 |
| | $\beta^*$ | 0.138 | -0.205 | 0.025 | -0.007 |
| | t-value | 44.802 | -67.761 | 8.182 | -2.158 |
| | p-value | 0.000 | 0.000 | 0.000 | 0.031 |
| PET_mean | $R^2$ | 0.040 | 0.066 | 0.012 | 0.001 |
| | $\beta^*$ | 0.280 | -0.450 | 0.130 | 0.035 |
| | t-value | 47.363 | -73.856 | 23.794 | 5.477 |
| | p-value | 0.000 | 0.000 | 0.000 | 0.000 |
| PET_sd | $R^2$ | 0.004 | 0.006 | -0.000 | 0.000 |
| | $\beta^*$ | 0.036 | -0.054 | -0.000 | -0.005 |
| | t-value | 13.124 | -20.012 | -0.010 | -1.901 |
| | p-value | 0.000 | 0.000 | 0.992 | 0.057 |
| SM_mean | $R^2$ | 0.032 | 0.014 | 0.010 | 0.001 |
| | $\beta^*$ | -0.249 | 0.185 | -0.159 | 0.038 |
| | t-value | -29.224 | 20.918 | -18.552 | 4.174 |
| | p-value | 0.000 | 0.000 | 0.000 | 0.000 |
| SM_sd | $R^2$ | 0.002 | 0.000 | 0.000 | 0.000 |
| | $\beta^*$ | 0.053 | -0.024 | -0.009 | 0.000 |
| | t-value | 8.204 | -3.576 | -1.402 | 0.026 |
| | p-value | 0.000 | 0.000 | 0.161 | 0.979 |
| VPD_mean | $R^2$ | 0.223 | 0.143 | 0.016 | 0.080 |
| | $\beta^*$ | 0.752 | -0.761 | 0.192 | -0.446 |
| | t-value | 111.805 | -107.803 | 25.697 | -80.869 |
| | p-value | 0.000 | 0.000 | 0.000 | 0.000 |
| VPD_sd | $R^2$ | 0.229 | 0.190 | 0.030 | 0.045 |
| | $\beta^*$ | 0.738 | -0.783 | 0.300 | -0.368 |
| | t-value | 100.753 | -85.300 | 30.734 | -33.522 |
| | p-value | 0.000 | 0.000 | 0.000 | 0.000 |
| LSTday_mean | $R^2$ | 0.013 | 0.064 | 0.003 | 0.000 |
| | $\beta^*$ | 0.130 | -0.363 | -0.037 | 0.001 |
| | t-value | 57.149 | -166.067 | -20.451 | 0.356 |
| | p-value | 0.000 | 0.000 | 0.000 | 0.722 |
| LSTday_sd | $R^2$ | 0.052 | 0.098 | 0.003 | 0.017 |
| | $\beta^*$ | 0.269 | -0.532 | -0.041 | -0.083 |
| | t-value | 83.279 | -161.036 | -14.199 | -29.066 |
| | p-value | 0.000 | 0.000 | 0.000 | 0.000 |
| LSTnight_mean | $R^2$ | 0.021 | 0.008 | 0.000 | 0.000 |
| | $\beta^*$ | 0.188 | -0.131 | -0.008 | 0.017 |
| | t-value | 54.450 | -36.133 | -2.296 | 4.865 |
| | p-value | 0.000 | 0.000 | 0.022 | 0.000 |
| LSTnight_sd | $R^2$ | 0.027 | 0.012 | 0.022 | 0.000 |
| | $\beta^*$ | -0.209 | 0.166 | -0.227 | 0.024 |
| | t-value | -30.274 | 24.131 | -32.270 | 3.483 |
| | p-value | 0.000 | 0.000 | 0.000 | 0.000 |





**Table A13.** Quantile regression results (Q50) for the Solimoes Sub-basin

| PARAMETER | STAT | SLA | LDMC | LNC | LPC |
|---|---|---|---|---|---|
| ET_mean | $R^2$ | 0.017 | 0.092 | 0.011 | 0.005 |
| | $\beta^*$ | -0.149 | 0.607 | 0.112 | -0.049 |
| | t-value | -103.872 | 400.161 | 74.347 | -34.630 |
| | p-value | 0.000 | 0.000 | 0.000 | 0.000 |
| ET_sd | $R^2$ | 0.004 | 0.008 | 0.001 | 0.014 |
| | $\beta^*$ | -0.060 | 0.079 | -0.044 | -0.102 |
| | t-value | -25.266 | 32.846 | -18.917 | -43.213 |
| | p-value | 0.000 | 0.000 | 0.000 | 0.000 |
| PET_mean | $R^2$ | 0.000 | 0.007 | 0.000 | 0.075 |
| | $\beta^*$ | -0.030 | 0.116 | 0.012 | -0.296 |
| | t-value | -8.135 | 29.823 | 3.120 | -116.914 |
| | p-value | 0.000 | 0.000 | 0.002 | 0.000 |
| PET_sd | $R^2$ | 0.021 | 0.036 | 0.002 | 0.064 |
| | $\beta^*$ | -0.163 | 0.223 | -0.060 | -0.235 |
| | t-value | -44.564 | 64.726 | -15.630 | -86.667 |
| | p-value | 0.000 | 0.000 | 0.000 | 0.000 |
| SM_mean | $R^2$ | 0.052 | 0.072 | 0.000 | 0.006 |
| | $\beta^*$ | -0.414 | 0.509 | -0.032 | 0.125 |
| | t-value | -82.584 | 102.238 | -5.859 | 23.737 |
| | p-value | 0.000 | 0.000 | 0.000 | 0.000 |
| SM_sd | $R^2$ | 0.004 | 0.001 | 0.000 | 0.027 |
| | $\beta^*$ | 0.076 | -0.040 | 0.011 | -0.220 |
| | t-value | 16.253 | -8.747 | 2.310 | -47.373 |
| | p-value | 0.000 | 0.000 | 0.021 | 0.000 |
| VPD_mean | $R^2$ | 0.000 | 0.007 | 0.000 | 0.001 |
| | $\beta^*$ | 0.010 | 0.119 | -0.008 | -0.023 |
| | t-value | 4.624 | 54.056 | -3.901 | -10.813 |
| | p-value | 0.000 | 0.000 | 0.000 | 0.000 |
| VPD_sd | $R^2$ | 0.000 | 0.001 | 0.000 | 0.153 |
| | $\beta^*$ | -0.025 | 0.060 | -0.023 | -0.524 |
| | t-value | -5.628 | 13.602 | -5.014 | -141.476 |
| | p-value | 0.000 | 0.000 | 0.000 | 0.000 |
| LSTday_mean | $R^2$ | 0.011 | 0.050 | 0.003 | 0.000 |
| | $\beta^*$ | -0.086 | 0.283 | 0.039 | -0.011 |
| | t-value | -129.728 | 455.407 | 50.041 | -13.491 |
| | p-value | 0.000 | 0.000 | 0.000 | 0.000 |
| LSTday_sd | $R^2$ | 0.075 | 0.151 | 0.000 | 0.000 |
| | $\beta^*$ | 0.249 | -0.683 | -0.003 | 0.003 |
| | t-value | 202.793 | -533.405 | -2.363 | 2.251 |
| | p-value | 0.000 | 0.000 | 0.018 | 0.024 |
| LSTnight_mean | $R^2$ | 0.016 | 0.076 | 0.000 | 0.004 |
| | $\beta^*$ | -0.078 | 0.421 | 0.013 | 0.025 |
| | t-value | -127.109 | 694.045 | 24.182 | 50.835 |
| | p-value | 0.000 | 0.000 | 0.000 | 0.000 |
| LSTnight_sd | $R^2$ | 0.001 | 0.000 | 0.002 | 0.012 |
| | $\beta^*$ | -0.025 | 0.011 | -0.054 | -0.123 |
| | t-value | -8.501 | 3.646 | -18.553 | -44.440 |
| | p-value | 0.000 | 0.000 | 0.000 | 0.000 |





**Table A14.** Quantile regression results (Q50) for the Tapajos Sub-basin

| PARAMETER | STAT | SLA | LDMC | LNC | LPC |
|---|---|---|---|---|---|
| ET_mean | $R^2$ | 0.146 | 0.122 | 0.001 | 0.115 |
| | $\beta^*$ | -0.523 | 0.437 | -0.027 | -0.295 |
| | t-value | -126.988 | 101.352 | -6.384 | -84.081 |
| | p-value | 0.000 | 0.000 | 0.000 | 0.000 |
| ET_sd | $R^2$ | 0.078 | 0.069 | 0.002 | 0.055 |
| | $\beta^*$ | 0.386 | -0.349 | 0.057 | 0.266 |
| | t-value | 54.408 | -48.543 | 8.817 | 39.946 |
| | p-value | 0.000 | 0.000 | 0.000 | 0.000 |
| PET_mean | $R^2$ | 0.131 | 0.133 | 0.001 | 0.174 |
| | $\beta^*$ | 0.505 | -0.529 | 0.040 | 0.530 |
| | t-value | 70.643 | -76.558 | 5.245 | 77.279 |
| | p-value | 0.000 | 0.000 | 0.000 | 0.000 |
| PET_sd | $R^2$ | 0.086 | 0.083 | 0.001 | 0.097 |
| | $\beta^*$ | 0.372 | -0.371 | 0.034 | 0.355 |
| | t-value | 45.270 | -47.553 | 4.189 | 53.285 |
| | p-value | 0.000 | 0.000 | 0.000 | 0.000 |
| SM_mean | $R^2$ | 0.028 | 0.024 | 0.000 | 0.031 |
| | $\beta^*$ | 0.206 | -0.202 | 0.005 | 0.213 |
| | t-value | 26.292 | -26.437 | 0.612 | 26.797 |
| | p-value | 0.000 | 0.000 | 0.540 | 0.000 |
| SM_sd | $R^2$ | 0.001 | 0.001 | 0.000 | 0.000 |
| | $\beta^*$ | 0.044 | -0.040 | 0.015 | 0.026 |
| | t-value | 4.700 | -4.365 | 1.695 | 2.765 |
| | p-value | 0.000 | 0.000 | 0.090 | 0.006 |
| VPD_mean | $R^2$ | 0.017 | 0.015 | 0.009 | 0.006 |
| | $\beta^*$ | 0.127 | -0.142 | 0.124 | 0.084 |
| | t-value | 19.959 | -22.931 | 18.547 | 13.616 |
| | p-value | 0.000 | 0.000 | 0.000 | 0.000 |
| VPD_sd | $R^2$ | 0.002 | 0.005 | 0.002 | 0.006 |
| | $\beta^*$ | 0.050 | -0.088 | 0.057 | 0.083 |
| | t-value | 8.054 | -13.935 | 9.249 | 13.452 |
| | p-value | 0.000 | 0.000 | 0.000 | 0.000 |
| LSTday_mean | $R^2$ | 0.296 | 0.297 | 0.001 | 0.255 |
| | $\beta^*$ | 0.701 | -0.715 | 0.070 | 0.595 |
| | t-value | 123.734 | -129.944 | 5.952 | 91.282 |
| | p-value | 0.000 | 0.000 | 0.000 | 0.000 |
| LSTday_sd | $R^2$ | 0.231 | 0.239 | 0.000 | 0.217 |
| | $\beta^*$ | 0.532 | -0.556 | 0.025 | 0.479 |
| | t-value | 103.721 | -115.513 | 2.767 | 87.242 |
| | p-value | 0.000 | 0.000 | 0.006 | 0.000 |
| LSTnight_mean | $R^2$ | 0.084 | 0.093 | 0.000 | 0.169 |
| | $\beta^*$ | -0.331 | 0.392 | 0.001 | -0.475 |
| | t-value | -37.829 | 44.250 | 0.083 | -75.356 |
| | p-value | 0.000 | 0.000 | 0.933 | 0.000 |
| LSTnight_sd | $R^2$ | 0.028 | 0.023 | 0.006 | 0.026 |
| | $\beta^*$ | 0.240 | -0.229 | -0.128 | 0.220 |
| | t-value | 24.797 | -23.057 | -12.810 | 22.417 |
| | p-value | 0.000 | 0.000 | 0.000 | 0.000 |



**Table A15.** Quantile regression results (Q50) for the Trombetas Sub-basin

| PARAMETER | STAT | SLA | LDMC | LNC | LPC |
|---|---|---|---|---|---|
| ET_mean | $R^2$ | 0.003 | 0.003 | 0.013 | 0.033 |
| | $\beta^*$ | -0.066 | 0.088 | 0.160 | -0.174 |
| | t-value | -9.940 | 13.051 | 24.047 | -28.013 |
| | p-value | 0.000 | 0.000 | 0.000 | 0.000 |
| ET_sd | $R^2$ | 0.006 | 0.004 | 0.004 | 0.002 |
| | $\beta^*$ | 0.107 | -0.111 | -0.099 | 0.054 |
| | t-value | 10.852 | -11.404 | -9.989 | 5.591 |
| | p-value | 0.000 | 0.000 | 0.000 | 0.000 |
| PET_mean | $R^2$ | 0.000 | 0.002 | 0.005 | 0.086 |
| | $\beta^*$ | -0.022 | 0.066 | 0.095 | -0.409 |
| | t-value | -1.775 | 5.148 | 7.623 | -47.289 |
| | p-value | 0.076 | 0.000 | 0.000 | 0.000 |
| PET_sd | $R^2$ | 0.005 | 0.001 | 0.010 | 0.007 |
| | $\beta^*$ | 0.124 | -0.056 | -0.184 | -0.114 |
| | t-value | 8.793 | -3.805 | -12.864 | -7.863 |
| | p-value | 0.000 | 0.000 | 0.000 | 0.000 |
| SM_mean | $R^2$ | 0.001 | 0.000 | 0.004 | 0.033 |
| | $\beta^*$ | 0.036 | 0.000 | -0.096 | 0.285 |
| | t-value | 3.175 | 0.015 | -8.554 | 25.608 |
| | p-value | 0.002 | 0.988 | 0.000 | 0.000 |
| SM_sd | $R^2$ | 0.002 | 0.001 | 0.001 | 0.004 |
| | $\beta^*$ | -0.057 | 0.044 | 0.046 | -0.079 |
| | t-value | -6.208 | 4.697 | 4.949 | -8.340 |
| | p-value | 0.000 | 0.000 | 0.000 | 0.000 |
| VPD_mean | $R^2$ | 0.002 | 0.000 | 0.000 | 0.037 |
| | $\beta^*$ | -0.069 | 0.010 | 0.018 | -0.346 |
| | t-value | -5.188 | 0.779 | 1.360 | -24.006 |
| | p-value | 0.000 | 0.436 | 0.174 | 0.000 |
| VPD_sd | $R^2$ | 0.028 | 0.019 | 0.005 | 0.053 |
| | $\beta^*$ | 0.215 | -0.289 | 0.143 | -0.385 |
| | t-value | 22.651 | -31.526 | 14.442 | -37.493 |
| | p-value | 0.000 | 0.000 | 0.000 | 0.000 |
| LSTday_mean | $R^2$ | 0.014 | 0.001 | 0.000 | 0.012 |
| | $\beta^*$ | 0.181 | 0.046 | 0.001 | -0.118 |
| | t-value | 25.927 | 6.846 | 0.212 | -18.257 |
| | p-value | 0.000 | 0.000 | 0.832 | 0.000 |
| LSTday_sd | $R^2$ | 0.043 | 0.022 | 0.006 | 0.008 |
| | $\beta^*$ | 0.338 | -0.253 | -0.085 | 0.085 |
| | t-value | 52.518 | -42.144 | -14.106 | 13.589 |
| | p-value | 0.000 | 0.000 | 0.000 | 0.000 |
| LSTnight_mean | $R^2$ | 0.000 | 0.010 | 0.030 | 0.012 |
| | $\beta^*$ | 0.026 | 0.287 | -0.276 | 0.084 |
| | t-value | 4.913 | 55.228 | -49.716 | 15.921 |
| | p-value | 0.000 | 0.000 | 0.000 | 0.000 |
| LSTnight_sd | $R^2$ | 0.004 | 0.007 | 0.012 | 0.004 |
| | $\beta^*$ | -0.090 | 0.192 | -0.198 | -0.081 |
| | t-value | -8.654 | 18.605 | -19.483 | -8.083 |
| | p-value | 0.000 | 0.000 | 0.000 | 0.000 |



**Table A16.** Quantile regression results (Q50) for the Xingu Sub-basin

| PARAMETER | STAT | SLA | LDMC | LNC | LPC |
|---|---|---|---|---|---|
| ET_mean | $R^2$ | 0.069 | 0.053 | 0.018 | 0.145 |
| | $\beta^*$ | -0.355 | 0.336 | 0.211 | -0.411 |
| | t-value | -49.090 | 48.650 | 32.122 | -90.873 |
| | p-value | 0.000 | 0.000 | 0.000 | 0.000 |
| ET_sd | $R^2$ | 0.033 | 0.018 | 0.002 | 0.102 |
| | $\beta^*$ | 0.262 | -0.234 | -0.097 | 0.442 |
| | t-value | 25.923 | -23.916 | -9.911 | 53.418 |
| | p-value | 0.000 | 0.000 | 0.000 | 0.000 |
| PET_mean | $R^2$ | 0.031 | 0.017 | 0.003 | 0.121 |
| | $\beta^*$ | 0.202 | -0.160 | -0.088 | 0.444 |
| | t-value | 33.115 | -30.265 | -16.721 | 70.501 |
| | p-value | 0.000 | 0.000 | 0.000 | 0.000 |
| PET_sd | $R^2$ | 0.008 | 0.001 | 0.002 | 0.060 |
| | $\beta^*$ | 0.148 | -0.044 | 0.048 | 0.345 |
| | t-value | 13.362 | -4.137 | 4.631 | 37.849 |
| | p-value | 0.000 | 0.000 | 0.000 | 0.000 |
| SM_mean | $R^2$ | 0.002 | 0.001 | 0.000 | 0.000 |
| | $\beta^*$ | 0.059 | -0.037 | -0.008 | 0.010 |
| | t-value | 6.162 | -3.801 | -0.857 | 0.972 |
| | p-value | 0.000 | 0.000 | 0.391 | 0.331 |
| SM_sd | $R^2$ | 0.001 | 0.000 | 0.000 | 0.000 |
| | $\beta^*$ | 0.028 | -0.021 | 0.005 | 0.019 |
| | t-value | 3.619 | -2.702 | 0.643 | 2.521 |
| | p-value | 0.000 | 0.007 | 0.520 | 0.012 |
| VPD_mean | $R^2$ | 0.014 | 0.001 | 0.003 | 0.055 |
| | $\beta^*$ | 0.219 | -0.074 | 0.066 | 0.375 |
| | t-value | 15.241 | -4.600 | 3.906 | 29.171 |
| | p-value | 0.000 | 0.000 | 0.000 | 0.000 |
| VPD_sd | $R^2$ | 0.008 | 0.001 | 0.001 | 0.071 |
| | $\beta^*$ | 0.129 | -0.070 | 0.050 | 0.354 |
| | t-value | 11.643 | -4.994 | 3.428 | 33.908 |
| | p-value | 0.000 | 0.000 | 0.001 | 0.000 |
| LSTday_mean | $R^2$ | 0.183 | 0.139 | 0.004 | 0.195 |
| | $\beta^*$ | 0.622 | -0.597 | -0.098 | 0.546 |
| | t-value | 96.851 | -102.904 | -15.508 | 87.040 |
| | p-value | 0.000 | 0.000 | 0.000 | 0.000 |
| LSTday_sd | $R^2$ | 0.281 | 0.242 | 0.002 | 0.133 |
| | $\beta^*$ | 0.700 | -0.724 | -0.055 | 0.508 |
| | t-value | 146.312 | -146.758 | -8.500 | 70.167 |
| | p-value | 0.000 | 0.000 | 0.000 | 0.000 |
| LSTnight_mean | $R^2$ | 0.044 | 0.028 | 0.000 | 0.154 |
| | $\beta^*$ | -0.312 | 0.307 | 0.001 | -0.500 |
| | t-value | -26.804 | 26.006 | 0.087 | -69.838 |
| | p-value | 0.000 | 0.000 | 0.931 | 0.000 |
| LSTnight_sd | $R^2$ | 0.065 | 0.051 | 0.020 | 0.050 |
| | $\beta^*$ | 0.390 | -0.385 | -0.285 | 0.343 |
| | t-value | 43.064 | -41.084 | -30.218 | 35.606 |
| | p-value | 0.000 | 0.000 | 0.000 | 0.000 |





**Table A17.** Quantile regression results (Q95) for the Amazon Basin

| PARAMETER | STAT | SLA | LDMC | LNC | LPC |
|---|---|---|---|---|---|
| ET_mean | $R^2$ | 0.003 | 0.000 | 0.003 | 0.010 |
| | $\beta^*$ | -0.038 | 0.013 | 0.027 | -0.049 |
| | t-value | -40.949 | 10.585 | 15.438 | -48.376 |
| | p-value | 0.000 | 0.000 | 0.000 | 0.000 |
| ET_sd | $R^2$ | 0.050 | 0.029 | 0.002 | 0.004 |
| | $\beta^*$ | 0.406 | -0.534 | -0.146 | -0.210 |
| | t-value | 42.654 | -86.136 | -17.898 | -25.055 |
| | p-value | 0.000 | 0.000 | 0.000 | 0.000 |
| PET_mean | $R^2$ | 0.182 | 0.121 | 0.005 | 0.028 |
| | $\beta^*$ | 0.492 | -0.704 | 0.103 | -0.378 |
| | t-value | 117.982 | -261.845 | 8.059 | -113.454 |
| | p-value | 0.000 | 0.000 | 0.000 | 0.000 |
| PET_sd | $R^2$ | 0.020 | 0.013 | 0.003 | 0.000 |
| | $\beta^*$ | 0.193 | -0.225 | -0.144 | 0.033 |
| | t-value | 41.410 | -64.183 | -35.195 | 7.514 |
| | p-value | 0.000 | 0.000 | 0.000 | 0.000 |
| SM_mean | $R^2$ | 0.042 | 0.012 | 0.013 | 0.062 |
| | $\beta^*$ | -0.277 | 0.129 | -0.174 | 0.293 |
| | t-value | -70.070 | 19.004 | -74.903 | 69.865 |
| | p-value | 0.000 | 0.000 | 0.000 | 0.000 |
| SM_sd | $R^2$ | 0.000 | 0.000 | 0.005 | 0.000 |
| | $\beta^*$ | -0.010 | 0.001 | -0.173 | -0.004 |
| | t-value | -1.285 | 0.071 | -26.915 | -0.600 |
| | p-value | 0.199 | 0.943 | 0.000 | 0.549 |
| VPD_mean | $R^2$ | 0.133 | 0.076 | 0.006 | 0.011 |
| | $\beta^*$ | 0.467 | -0.474 | 0.094 | -0.276 |
| | t-value | 86.256 | -118.690 | 6.566 | -44.230 |
| | p-value | 0.000 | 0.000 | 0.000 | 0.000 |
| VPD_sd | $R^2$ | 0.027 | 0.017 | 0.007 | 0.034 |
| | $\beta^*$ | 0.317 | -0.367 | 0.109 | -0.555 |
| | t-value | 30.121 | -90.892 | 11.066 | -144.323 |
| | p-value | 0.000 | 0.000 | 0.000 | 0.000 |
| LSTday_mean | $R^2$ | 0.264 | 0.279 | 0.004 | 0.017 |
| | $\beta^*$ | 0.533 | -0.706 | 0.067 | 0.228 |
| | t-value | 112.075 | -171.753 | 5.638 | 34.107 |
| | p-value | 0.000 | 0.000 | 0.000 | 0.000 |
| LSTday_sd | $R^2$ | 0.179 | 0.263 | 0.002 | 0.121 |
| | $\beta^*$ | 1.002 | -0.969 | 0.068 | 0.749 |
| | t-value | 67.346 | -83.839 | 2.395 | 56.191 |
| | p-value | 0.000 | 0.000 | 0.017 | 0.000 |
| LSTnight_mean | $R^2$ | 0.006 | 0.003 | 0.041 | 0.008 |
| | $\beta^*$ | -0.051 | -0.033 | -0.120 | 0.052 |
| | t-value | -28.519 | -14.237 | -95.261 | 31.135 |
| | p-value | 0.000 | 0.000 | 0.000 | 0.000 |
| LSTnight_sd | $R^2$ | 0.044 | 0.048 | 0.000 | 0.001 |
| | $\beta^*$ | 0.390 | -0.580 | 0.014 | 0.074 |
| | t-value | 31.228 | -74.178 | 1.285 | 9.178 |
| | p-value | 0.000 | 0.000 | 0.199 | 0.000 |




**Table A18.** Quantile regression results (Q95) for the Amazonas Sub-basin

| PARAMETER | STAT | SLA | LDMC | LNC | LPC |
|---|---|---|---|---|---|
| ET_mean | $R^2$ | 0.007 | 0.010 | 0.002 | 0.000 |
| | $\beta^*$ | -0.066 | 0.050 | 0.045 | 0.002 |
| | t-value | -10.071 | 3.379 | 4.911 | 0.294 |
| | p-value | 0.000 | 0.001 | 0.000 | 0.769 |
| ET_sd | $R^2$ | 0.059 | 0.031 | 0.001 | 0.007 |
| | $\beta^*$ | 0.407 | -0.544 | 0.031 | 0.122 |
| | t-value | 6.675 | -15.592 | 0.502 | 1.741 |
| | p-value | 0.000 | 0.000 | 0.615 | 0.082 |
| PET_mean | $R^2$ | 0.000 | 0.009 | 0.006 | 0.000 |
| | $\beta^*$ | -0.007 | 0.087 | 0.085 | 0.012 |
| | t-value | -0.907 | 5.810 | 6.030 | 1.281 |
| | p-value | 0.364 | 0.000 | 0.000 | 0.200 |
| PET_sd | $R^2$ | 0.014 | 0.000 | 0.002 | 0.000 |
| | $\beta^*$ | 0.206 | -0.013 | 0.030 | -0.016 |
| | t-value | 3.414 | -0.415 | 0.330 | -0.454 |
| | p-value | 0.001 | 0.678 | 0.741 | 0.650 |
| SM_mean | $R^2$ | 0.001 | 0.020 | 0.031 | 0.001 |
| | $\beta^*$ | 0.031 | -0.298 | -0.366 | -0.033 |
| | t-value | 0.604 | -9.878 | -14.440 | -0.580 |
| | p-value | 0.546 | 0.000 | 0.000 | 0.562 |
| SM_sd | $R^2$ | 0.004 | 0.019 | 0.019 | 0.002 |
| | $\beta^*$ | 0.152 | -0.409 | -0.359 | 0.103 |
| | t-value | 3.119 | -16.210 | -10.385 | 1.350 |
| | p-value | 0.002 | 0.000 | 0.000 | 0.177 |
| VPD_mean | $R^2$ | 0.008 | 0.003 | 0.000 | 0.008 |
| | $\beta^*$ | -0.088 | 0.037 | 0.014 | -0.088 |
| | t-value | -4.391 | 0.809 | 0.521 | -4.285 |
| | p-value | 0.000 | 0.418 | 0.602 | 0.000 |
| VPD_sd | $R^2$ | 0.081 | 0.051 | 0.003 | 0.028 |
| | $\beta^*$ | 0.255 | -0.345 | 0.026 | 0.253 |
| | t-value | 10.832 | -24.352 | 0.400 | 3.278 |
| | p-value | 0.000 | 0.000 | 0.689 | 0.001 |
| LSTday_mean | $R^2$ | 0.235 | 0.175 | 0.006 | 0.082 |
| | $\beta^*$ | 0.674 | -1.033 | 0.069 | 0.564 |
| | t-value | 12.110 | -32.373 | 0.566 | 5.975 |
| | p-value | 0.000 | 0.000 | 0.571 | 0.000 |
| LSTday_sd | $R^2$ | 0.281 | 0.261 | 0.000 | 0.065 |
| | $\beta^*$ | 1.057 | -1.580 | -0.004 | 0.742 |
| | t-value | 15.472 | -40.205 | -0.030 | 4.238 |
| | p-value | 0.000 | 0.000 | 0.976 | 0.000 |
| LSTnight_mean | $R^2$ | 0.006 | 0.137 | 0.183 | 0.000 |
| | $\beta^*$ | 0.221 | -1.135 | -0.844 | -0.040 |
| | t-value | 1.994 | -30.365 | -28.776 | -0.275 |
| | p-value | 0.046 | 0.000 | 0.000 | 0.783 |
| LSTnight_sd | $R^2$ | 0.000 | 0.004 | 0.022 | 0.002 |
| | $\beta^*$ | 0.005 | -0.146 | -0.368 | 0.050 |
| | t-value | 0.119 | -4.876 | -13.507 | 0.742 |
| | p-value | 0.905 | 0.000 | 0.000 | 0.458 |





**Table A19.** Quantile regression results (Q95) for the Madeira Sub-basin

| PARAMETER | STAT | SLA | LDMC | LNC | LPC |
|---|---|---|---|---|---|
| ET_mean | $R^2$ | 0.011 | 0.004 | 0.000 | 0.010 |
| | $\beta^*$ | -0.040 | 0.024 | 0.010 | -0.033 |
| | t-value | -28.803 | 13.526 | 8.097 | -23.984 |
| | p-value | 0.000 | 0.000 | 0.000 | 0.000 |
| ET_sd | $R^2$ | 0.029 | 0.012 | 0.000 | 0.001 |
| | $\beta^*$ | 0.339 | -0.390 | -0.004 | -0.059 |
| | t-value | 25.916 | -53.200 | -0.409 | -6.591 |
| | p-value | 0.000 | 0.000 | 0.682 | 0.000 |
| PET_mean | $R^2$ | 0.070 | 0.040 | 0.002 | 0.003 |
| | $\beta^*$ | 0.389 | -0.447 | 0.033 | -0.102 |
| | t-value | 48.819 | -137.236 | 3.528 | -29.001 |
| | p-value | 0.000 | 0.000 | 0.000 | 0.000 |
| PET_sd | $R^2$ | 0.000 | 0.002 | 0.004 | 0.004 |
| | $\beta^*$ | 0.043 | -0.121 | -0.176 | 0.146 |
| | t-value | 3.136 | -9.620 | -13.138 | 8.852 |
| | p-value | 0.002 | 0.000 | 0.000 | 0.000 |
| SM_mean | $R^2$ | 0.039 | 0.013 | 0.000 | 0.003 |
| | $\beta^*$ | -0.238 | 0.114 | 0.014 | -0.070 |
| | t-value | -38.421 | 8.481 | 3.110 | -11.490 |
| | p-value | 0.000 | 0.000 | 0.002 | 0.000 |
| SM_sd | $R^2$ | 0.000 | 0.001 | 0.000 | 0.001 |
| | $\beta^*$ | 0.005 | -0.075 | -0.053 | -0.064 |
| | t-value | 0.378 | -5.136 | -3.444 | -5.131 |
| | p-value | 0.705 | 0.000 | 0.001 | 0.000 |
| VPD_mean | $R^2$ | 0.124 | 0.053 | 0.004 | 0.027 |
| | $\beta^*$ | 0.667 | -0.787 | 0.075 | -0.352 |
| | t-value | 42.056 | -76.507 | 2.359 | -24.305 |
| | p-value | 0.000 | 0.000 | 0.018 | 0.000 |
| VPD_sd | $R^2$ | 0.080 | 0.016 | 0.012 | 0.032 |
| | $\beta^*$ | 0.267 | -0.225 | 0.083 | -0.199 |
| | t-value | 79.607 | -95.770 | 6.135 | -62.423 |
| | p-value | 0.000 | 0.000 | 0.000 | 0.000 |
| LSTday_mean | $R^2$ | 0.118 | 0.141 | 0.002 | 0.045 |
| | $\beta^*$ | 0.321 | -0.513 | 0.029 | 0.218 |
| | t-value | 36.860 | -94.583 | 2.333 | 24.157 |
| | p-value | 0.000 | 0.000 | 0.020 | 0.000 |
| LSTday_sd | $R^2$ | 0.168 | 0.161 | 0.000 | 0.144 |
| | $\beta^*$ | 0.817 | -0.824 | -0.006 | 0.581 |
| | t-value | 51.850 | -77.167 | -0.249 | 31.274 |
| | p-value | 0.000 | 0.000 | 0.804 | 0.000 |
| LSTnight_mean | $R^2$ | 0.032 | 0.004 | 0.011 | 0.003 |
| | $\beta^*$ | -0.079 | 0.023 | -0.058 | 0.023 |
| | t-value | -40.146 | 5.112 | -32.311 | 10.558 |
| | p-value | 0.000 | 0.000 | 0.000 | 0.000 |
| LSTnight_sd | $R^2$ | 0.035 | 0.016 | 0.000 | 0.023 |
| | $\beta^*$ | 0.319 | -0.310 | -0.025 | 0.251 |
| | t-value | 22.585 | -29.629 | -1.991 | 14.667 |
| | p-value | 0.000 | 0.000 | 0.047 | 0.000 |



**Table A20.** Quantile regression results (Q95) for the Negro Sub-basin

| PARAMETER | STAT | SLA | LDMC | LNC | LPC |
|---|---|---|---|---|---|
| ET_mean | $R^2$ | 0.041 | 0.015 | 0.021 | 0.002 |
| | $\beta^*$ | 0.175 | -0.170 | 0.107 | -0.037 |
| | t-value | 23.277 | -34.701 | 8.355 | -5.241 |
| | p-value | 0.000 | 0.000 | 0.000 | 0.000 |
| ET_sd | $R^2$ | 0.168 | 0.214 | 0.000 | 0.000 |
| | $\beta^*$ | 0.498 | -0.595 | 0.001 | -0.023 |
| | t-value | 27.266 | -34.023 | 0.033 | -1.042 |
| | p-value | 0.000 | 0.000 | 0.974 | 0.297 |
| PET_mean | $R^2$ | 0.300 | 0.339 | 0.006 | 0.007 |
| | $\beta^*$ | 0.878 | -1.039 | 0.153 | -0.390 |
| | t-value | 31.068 | -55.476 | 1.448 | -9.444 |
| | p-value | 0.000 | 0.000 | 0.148 | 0.000 |
| PET_sd | $R^2$ | 0.022 | 0.030 | 0.002 | 0.000 |
| | $\beta^*$ | 0.144 | -0.210 | -0.044 | 0.031 |
| | t-value | 11.364 | -18.434 | -4.394 | 3.554 |
| | p-value | 0.000 | 0.000 | 0.000 | 0.000 |
| SM_mean | $R^2$ | 0.032 | 0.009 | 0.008 | 0.005 |
| | $\beta^*$ | -0.161 | 0.071 | -0.075 | 0.055 |
| | t-value | -27.560 | 7.326 | -18.744 | 8.138 |
| | p-value | 0.000 | 0.000 | 0.000 | 0.000 |
| SM_sd | $R^2$ | 0.025 | 0.011 | 0.025 | 0.010 |
| | $\beta^*$ | -0.332 | 0.168 | -0.419 | 0.182 |
| | t-value | -10.757 | 3.032 | -17.740 | 4.931 |
| | p-value | 0.000 | 0.002 | 0.000 | 0.000 |
| VPD_mean | $R^2$ | 0.328 | 0.370 | 0.018 | 0.002 |
| | $\beta^*$ | 0.690 | -0.667 | 0.181 | -0.121 |
| | t-value | 38.374 | -68.398 | 3.570 | -5.592 |
| | p-value | 0.000 | 0.000 | 0.000 | 0.000 |
| VPD_sd | $R^2$ | 0.244 | 0.293 | 0.018 | 0.004 |
| | $\beta^*$ | 0.458 | -0.522 | 0.166 | -0.221 |
| | t-value | 28.645 | -66.247 | 4.350 | -16.500 |
| | p-value | 0.000 | 0.000 | 0.000 | 0.000 |
| LSTday_mean | $R^2$ | 0.460 | 0.492 | 0.010 | 0.005 |
| | $\beta^*$ | 0.986 | -0.967 | 0.203 | 0.224 |
| | t-value | 49.369 | -30.535 | 0.941 | 2.841 |
| | p-value | 0.000 | 0.000 | 0.347 | 0.005 |
| LSTday_sd | $R^2$ | 0.433 | 0.413 | 0.005 | 0.001 |
| | $\beta^*$ | 1.075 | -1.018 | 0.135 | 0.128 |
| | t-value | 33.656 | -20.660 | 0.666 | 1.695 |
| | p-value | 0.000 | 0.000 | 0.505 | 0.090 |
| LSTnight_mean | $R^2$ | 0.093 | 0.091 | 0.012 | 0.003 |
| | $\beta^*$ | 0.296 | -0.261 | -0.147 | 0.092 |
| | t-value | 9.521 | -9.518 | -4.145 | 3.822 |
| | p-value | 0.000 | 0.000 | 0.000 | 0.000 |
| LSTnight_sd | $R^2$ | 0.026 | 0.004 | 0.020 | 0.007 |
| | $\beta^*$ | -0.290 | 0.080 | -0.253 | -0.151 |
| | t-value | -14.903 | 2.539 | -15.423 | -8.649 |
| | p-value | 0.000 | 0.011 | 0.000 | 0.000 |





**Table A21.** Quantile regression results (Q95) for the Solimoes Sub-basin

| PARAMETER | STAT | SLA | LDMC | LNC | LPC |
|---|---|---|---|---|---|
| ET_mean | $R^2$ | 0.003 | 0.001 | 0.005 | 0.003 |
| | $\beta^*$ | 0.034 | 0.017 | 0.035 | 0.022 |
| | t-value | 17.639 | 7.642 | 10.724 | 10.958 |
| | p-value | 0.000 | 0.000 | 0.000 | 0.000 |
| ET_sd | $R^2$ | 0.000 | 0.000 | 0.065 | 0.005 |
| | $\beta^*$ | -0.055 | 0.022 | -0.913 | -0.229 |
| | t-value | -2.501 | 0.969 | -56.346 | -14.935 |
| | p-value | 0.012 | 0.333 | 0.000 | 0.000 |
| PET_mean | $R^2$ | 0.000 | 0.000 | 0.007 | 0.002 |
| | $\beta^*$ | -0.023 | 0.011 | -0.115 | -0.053 |
| | t-value | -5.265 | 2.258 | -36.687 | -14.496 |
| | p-value | 0.000 | 0.024 | 0.000 | 0.000 |
| PET_sd | $R^2$ | 0.000 | 0.001 | 0.029 | 0.006 |
| | $\beta^*$ | -0.030 | 0.070 | -0.653 | 0.127 |
| | t-value | -5.939 | 11.040 | -98.066 | 34.197 |
| | p-value | 0.000 | 0.000 | 0.000 | 0.000 |
| SM_mean | $R^2$ | 0.008 | 0.004 | 0.005 | 0.020 |
| | $\beta^*$ | -0.115 | 0.076 | -0.095 | 0.163 |
| | t-value | -33.717 | 14.217 | -41.750 | 43.778 |
| | p-value | 0.000 | 0.000 | 0.000 | 0.000 |
| SM_sd | $R^2$ | 0.001 | 0.000 | 0.004 | 0.003 |
| | $\beta^*$ | -0.040 | 0.029 | -0.159 | -0.109 |
| | t-value | -3.386 | 2.273 | -17.091 | -8.566 |
| | p-value | 0.001 | 0.023 | 0.000 | 0.000 |
| VPD_mean | $R^2$ | 0.012 | 0.002 | 0.000 | 0.006 |
| | $\beta^*$ | 0.178 | -0.119 | -0.004 | -0.145 |
| | t-value | 14.815 | -21.658 | -0.373 | -18.165 |
| | p-value | 0.000 | 0.000 | 0.709 | 0.000 |
| VPD_sd | $R^2$ | 0.012 | 0.002 | 0.000 | 0.101 |
| | $\beta^*$ | 0.130 | -0.114 | 0.026 | -0.498 |
| | t-value | 12.740 | -22.199 | 2.645 | -50.704 |
| | p-value | 0.000 | 0.000 | 0.008 | 0.000 |
| LSTday_mean | $R^2$ | 0.105 | 0.099 | 0.000 | 0.001 |
| | $\beta^*$ | 0.150 | -0.318 | 0.010 | -0.015 |
| | t-value | 38.661 | -267.302 | 1.782 | -4.574 |
| | p-value | 0.000 | 0.000 | 0.075 | 0.000 |
| LSTday_sd | $R^2$ | 0.215 | 0.374 | 0.004 | 0.255 |
| | $\beta^*$ | 0.923 | -1.073 | 0.116 | 1.099 |
| | t-value | 19.987 | -34.480 | 1.804 | 62.727 |
| | p-value | 0.000 | 0.000 | 0.071 | 0.000 |
| LSTnight_mean | $R^2$ | 0.008 | 0.000 | 0.016 | 0.016 |
| | $\beta^*$ | -0.040 | 0.002 | -0.047 | 0.050 |
| | t-value | -36.764 | 1.471 | -56.421 | 41.327 |
| | p-value | 0.000 | 0.141 | 0.000 | 0.000 |
| LSTnight_sd | $R^2$ | 0.007 | 0.004 | 0.022 | 0.091 |
| | $\beta^*$ | -0.207 | 0.128 | -0.549 | -0.710 |
| | t-value | -10.007 | 4.174 | -46.675 | -36.289 |
| | p-value | 0.000 | 0.000 | 0.000 | 0.000 |



**Table A22.** Quantile regression results (Q95) for the Tapajos Sub-basin

| PARAMETER | STAT | SLA | LDMC | LNC | LPC |
|---|---|---|---|---|---|
| ET_mean | $R^2$ | 0.036 | 0.029 | 0.001 | 0.015 |
| | $\beta^*$ | -0.114 | 0.101 | 0.008 | -0.055 |
| | t-value | -53.091 | 23.332 | 1.826 | -28.947 |
| | p-value | 0.000 | 0.000 | 0.068 | 0.000 |
| ET_sd | $R^2$ | 0.138 | 0.132 | 0.006 | 0.078 |
| | $\beta^*$ | 0.812 | -0.921 | 0.108 | 0.709 |
| | t-value | 28.741 | -36.224 | 1.925 | 18.923 |
| | p-value | 0.000 | 0.000 | 0.054 | 0.000 |
| PET_mean | $R^2$ | 0.125 | 0.183 | 0.002 | 0.149 |
| | $\beta^*$ | 0.664 | -0.831 | 0.080 | 0.702 |
| | t-value | 49.660 | -72.198 | 4.069 | 32.419 |
| | p-value | 0.000 | 0.000 | 0.000 | 0.000 |
| PET_sd | $R^2$ | 0.166 | 0.155 | 0.003 | 0.111 |
| | $\beta^*$ | 0.855 | -0.899 | 0.081 | 0.874 |
| | t-value | 31.169 | -33.279 | 1.481 | 25.944 |
| | p-value | 0.000 | 0.000 | 0.139 | 0.000 |
| SM_mean | $R^2$ | 0.082 | 0.066 | 0.005 | 0.051 |
| | $\beta^*$ | 0.487 | -0.502 | 0.084 | 0.416 |
| | t-value | 27.042 | -26.851 | 2.198 | 14.884 |
| | p-value | 0.000 | 0.000 | 0.028 | 0.000 |
| SM_sd | $R^2$ | 0.003 | 0.002 | 0.007 | 0.001 |
| | $\beta^*$ | -0.094 | 0.085 | -0.160 | -0.060 |
| | t-value | -4.033 | 3.380 | -8.094 | -2.711 |
| | p-value | 0.000 | 0.001 | 0.000 | 0.007 |
| VPD_mean | $R^2$ | 0.081 | 0.087 | 0.014 | 0.080 |
| | $\beta^*$ | 0.374 | -0.418 | 0.148 | 0.448 |
| | t-value | 21.987 | -27.227 | 2.688 | 18.160 |
| | p-value | 0.000 | 0.000 | 0.007 | 0.000 |
| VPD_sd | $R^2$ | 0.046 | 0.046 | 0.015 | 0.052 |
| | $\beta^*$ | 0.285 | -0.296 | 0.134 | 0.336 |
| | t-value | 18.871 | -19.890 | 2.868 | 17.874 |
| | p-value | 0.000 | 0.000 | 0.004 | 0.000 |
| LSTday_mean | $R^2$ | 0.132 | 0.162 | 0.003 | 0.111 |
| | $\beta^*$ | 0.640 | -0.796 | 0.065 | 0.656 |
| | t-value | 48.208 | -63.994 | 3.512 | 29.639 |
| | p-value | 0.000 | 0.000 | 0.000 | 0.000 |
| LSTday_sd | $R^2$ | 0.176 | 0.166 | 0.005 | 0.114 |
| | $\beta^*$ | 1.029 | -1.083 | 0.101 | 1.031 |
| | t-value | 42.143 | -44.327 | 1.175 | 29.273 |
| | p-value | 0.000 | 0.000 | 0.240 | 0.000 |
| LSTnight_mean | $R^2$ | 0.071 | 0.039 | 0.006 | 0.026 |
| | $\beta^*$ | -0.408 | 0.369 | -0.089 | -0.296 |
| | t-value | -44.703 | 17.110 | -6.546 | -30.582 |
| | p-value | 0.000 | 0.000 | 0.000 | 0.000 |
| LSTnight_sd | $R^2$ | 0.009 | 0.012 | 0.006 | 0.003 |
| | $\beta^*$ | 0.150 | -0.184 | -0.121 | 0.095 |
| | t-value | 6.542 | -9.159 | -5.622 | 3.405 |
| | p-value | 0.000 | 0.000 | 0.000 | 0.001 |



**Table A23.** Quantile regression results (Q95) for the Trombetas Sub-basin

| PARAMETER | STAT | SLA | LDMC | LNC | LPC |
|---|---|---|---|---|---|
| ET_mean | $R^2$ | 0.000 | 0.002 | 0.006 | 0.000 |
| | $\beta^*$ | -0.002 | 0.034 | 0.045 | -0.011 |
| | t-value | -0.500 | 3.601 | 4.279 | -1.478 |
| | p-value | 0.617 | 0.000 | 0.000 | 0.139 |
| ET_sd | $R^2$ | 0.008 | 0.021 | 0.019 | 0.004 |
| | $\beta^*$ | 0.117 | -0.272 | -0.325 | -0.144 |
| | t-value | 2.394 | -8.910 | -13.498 | -4.901 |
| | p-value | 0.017 | 0.000 | 0.000 | 0.000 |
| PET_mean | $R^2$ | 0.000 | 0.004 | 0.010 | 0.023 |
| | $\beta^*$ | -0.002 | 0.080 | -0.278 | -0.250 |
| | t-value | -0.250 | 4.118 | -54.319 | -30.003 |
| | p-value | 0.803 | 0.000 | 0.000 | 0.000 |
| PET_sd | $R^2$ | 0.002 | 0.010 | 0.032 | 0.013 |
| | $\beta^*$ | 0.054 | -0.113 | -0.312 | -0.183 |
| | t-value | 2.196 | -6.079 | -25.052 | -14.146 |
| | p-value | 0.028 | 0.000 | 0.000 | 0.000 |
| SM_mean | $R^2$ | 0.000 | 0.000 | 0.013 | 0.013 |
| | $\beta^*$ | -0.008 | 0.016 | -0.134 | 0.149 |
| | t-value | -0.404 | 0.514 | -9.794 | 8.004 |
| | p-value | 0.686 | 0.607 | 0.000 | 0.000 |
| SM_sd | $R^2$ | 0.003 | 0.005 | 0.005 | 0.054 |
| | $\beta^*$ | -0.186 | 0.103 | -0.233 | 0.394 |
| | t-value | -4.221 | 0.998 | -6.019 | 9.593 |
| | p-value | 0.000 | 0.318 | 0.000 | 0.000 |
| VPD_mean | $R^2$ | 0.044 | 0.024 | 0.005 | 0.100 |
| | $\beta^*$ | 0.197 | -0.313 | 0.082 | -0.428 |
| | t-value | 6.813 | -22.412 | 3.041 | -38.699 |
| | p-value | 0.000 | 0.000 | 0.002 | 0.000 |
| VPD_sd | $R^2$ | 0.000 | 0.001 | 0.001 | 0.135 |
| | $\beta^*$ | 0.012 | -0.043 | -0.047 | -0.399 |
| | t-value | 0.662 | -4.423 | -4.491 | -19.921 |
| | p-value | 0.508 | 0.000 | 0.000 | 0.000 |
| LSTday_mean | $R^2$ | 0.187 | 0.060 | 0.000 | 0.003 |
| | $\beta^*$ | 0.508 | -0.345 | 0.017 | 0.071 |
| | t-value | 14.012 | -9.902 | 0.275 | 2.423 |
| | p-value | 0.000 | 0.000 | 0.783 | 0.015 |
| LSTday_sd | $R^2$ | 0.145 | 0.061 | 0.004 | 0.000 |
| | $\beta^*$ | 0.705 | -0.463 | -0.088 | 0.037 |
| | t-value | 6.987 | -8.571 | -1.656 | 0.853 |
| | p-value | 0.000 | 0.000 | 0.098 | 0.394 |
| LSTnight_mean | $R^2$ | 0.003 | 0.003 | 0.194 | 0.006 |
| | $\beta^*$ | -0.094 | -0.061 | -0.792 | 0.228 |
| | t-value | -0.938 | -0.426 | -22.410 | 2.020 |
| | p-value | 0.348 | 0.670 | 0.000 | 0.043 |
| LSTnight_sd | $R^2$ | 0.000 | 0.004 | 0.020 | 0.022 |
| | $\beta^*$ | -0.005 | 0.067 | -0.270 | 0.204 |
| | t-value | -0.137 | 1.135 | -11.428 | 6.116 |
| | p-value | 0.891 | 0.256 | 0.000 | 0.000 |





**Table A24.** Quantile regression results (Q95) for the Xingu Sub-basin

| PARAMETER | STAT | SLA | LDMC | LNC | LPC |
|---|---|---|---|---|---|
| ET_mean | $R^2$ | 0.004 | 0.001 | 0.000 | 0.025 |
| | $\beta^*$ | -0.042 | 0.013 | 0.005 | -0.087 |
| | t-value | -16.540 | 2.895 | 1.209 | -28.550 |
| | p-value | 0.000 | 0.004 | 0.227 | 0.000 |
| ET_sd | $R^2$ | 0.088 | 0.077 | 0.001 | 0.066 |
| | $\beta^*$ | 0.438 | -0.466 | -0.058 | 0.432 |
| | t-value | 26.295 | -25.713 | -3.112 | 11.499 |
| | p-value | 0.000 | 0.000 | 0.002 | 0.000 |
| PET_mean | $R^2$ | 0.153 | 0.149 | 0.001 | 0.113 |
| | $\beta^*$ | 0.483 | -0.557 | 0.032 | 0.391 |
| | t-value | 33.312 | -52.221 | 4.051 | 13.514 |
| | p-value | 0.000 | 0.000 | 0.000 | 0.000 |
| PET_sd | $R^2$ | 0.121 | 0.111 | 0.001 | 0.103 |
| | $\beta^*$ | 0.566 | -0.614 | 0.084 | 0.595 |
| | t-value | 22.918 | -33.159 | 3.285 | 13.011 |
| | p-value | 0.000 | 0.000 | 0.001 | 0.000 |
| SM_mean | $R^2$ | 0.005 | 0.004 | 0.000 | 0.000 |
| | $\beta^*$ | 0.110 | -0.110 | 0.012 | 0.010 |
| | t-value | 6.049 | -6.674 | 0.688 | 0.485 |
| | p-value | 0.000 | 0.000 | 0.491 | 0.627 |
| SM_sd | $R^2$ | 0.001 | 0.003 | 0.004 | 0.000 |
| | $\beta^*$ | 0.096 | -0.115 | -0.170 | -0.025 |
| | t-value | 3.394 | -4.042 | -5.074 | -0.887 |
| | p-value | 0.001 | 0.000 | 0.000 | 0.375 |
| VPD_mean | $R^2$ | 0.001 | 0.002 | 0.019 | 0.019 |
| | $\beta^*$ | 0.044 | -0.048 | 0.118 | 0.183 |
| | t-value | 4.175 | -4.796 | 4.737 | 24.501 |
| | p-value | 0.000 | 0.000 | 0.000 | 0.000 |
| VPD_sd | $R^2$ | 0.000 | 0.002 | 0.015 | 0.017 |
| | $\beta^*$ | 0.019 | -0.050 | 0.119 | 0.195 |
| | t-value | 1.718 | -5.218 | 5.653 | 25.217 |
| | p-value | 0.086 | 0.000 | 0.000 | 0.000 |
| LSTday_mean | $R^2$ | 0.231 | 0.237 | 0.004 | 0.164 |
| | $\beta^*$ | 0.687 | -0.869 | 0.099 | 0.658 |
| | t-value | 34.359 | -53.812 | 4.180 | 17.227 |
| | p-value | 0.000 | 0.000 | 0.000 | 0.000 |
| LSTday_sd | $R^2$ | 0.203 | 0.197 | 0.000 | 0.108 |
| | $\beta^*$ | 0.761 | -0.925 | 0.023 | 0.597 |
| | t-value | 34.129 | -39.511 | 1.653 | 14.799 |
| | p-value | 0.000 | 0.000 | 0.098 | 0.000 |
| LSTnight_mean | $R^2$ | 0.005 | 0.004 | 0.080 | 0.034 |
| | $\beta^*$ | -0.116 | -0.088 | -0.476 | -0.295 |
| | t-value | -8.637 | -4.627 | -44.511 | -19.152 |
| | p-value | 0.000 | 0.000 | 0.000 | 0.000 |
| LSTnight_sd | $R^2$ | 0.053 | 0.059 | 0.007 | 0.030 |
| | $\beta^*$ | 0.334 | -0.427 | -0.146 | 0.265 |
| | t-value | 18.254 | -27.499 | -6.449 | 8.409 |
| | p-value | 0.000 | 0.000 | 0.000 | 0.000 |