# Peer review of "Effects of plant traits on the regulation of water cycle processes in the Amazon Basin"

_EGUsphere, 2024_

## Author Comment (AC1)

**Responses to comments from RC1**

**Manuscript: Effects of plant traits on the regulation of water cycle processes in the Amazon Basin**

Kien Nguyen and Maria J. Santos

Dear editor,

With this letter, we provide our reply to the comments provided by RC1 to our submitted manuscript entitled *"Effects of plant traits on the regulation of water cycle processes in the Amazon Basin"*. We appreciate the time that RC1 took to review our work and have seriously considered and appreciated all the comments by the reviewer. We detail in the following our plan to address the comments in a revised version.

**General Comments**

*The authors present a large number of quantile regression analyses modelling the linear relationships between modelled plant functional traits, and long-term averages of climate data across the Amazon basin, which they call 'parameters'. I apologise, as my comments will sound harsh and I can fully believe the lead author has done a lot of work to produce this. I would also like to state that I did read and attempt to comprehend the manuscript. However in my opinion, the manuscript is poorly organised, written, and ultimately very difficult to read and review. The abstract is too confusing for me to understand what the analysis was. The introduction is unfocused and loaded with long paragraphs containing generic statements about plant functional traits and vegetation hydrology. The methods are exceptionally brief and under detailed. Figure 1 is useful, but the other figures have serious presentation issues. Overall, I found it very difficult to comprehend what this manuscript was about, the justification for its analyses, and its results.*

**Response:** Thank you for your evaluation of our manuscript. We appreciate your perspective and the specific areas that you have highlighted for potential improvement. We acknowledge the importance of clarity and coherence in a manuscript. We will re-work the abstract and the introduction, based on your feedback. We will also add more detail in the Methods section and work in the figures. We hope our improved version will meet the standards you request. We value your input and your comments will be helpful as we work to further strengthen this manuscript.

**RC1.1.** *The problem statement premise of this manuscript appears to be: "however, detailed understanding on how feedbacks that involve plant traits develop are still seldom included in observational, experimental and modelling studies." but this is a strawman argument. The authors do not cite studies to support this argument. Many studies are cited, but there is no coherent narrative. It is a jumbled and superficial description of plant functional traits and vegetation hydrology.*

**Response:** Thank you for your comment. The idea behind the analysis is to show that while plant traits have been documented to be affected by environmental conditions, these same plant traits influence water exchanges. Models, experimental and observational studies have often considered either plant functional types or examined this effect at local scales (but see, for example, LPJGUESS Lm efforts to include plant traits). When previous studies include traits, these are often derived from field samples that are restricted in space and time (Jetz et al., 2016). Here, we examine whether it would be possible to use remote sensing-derived plant traits to fill the gap between modeling and in situ experimental and observational studies.

Moving forward, in a revised version, we will refine the Introduction, to ensure that the problem statement is clearly stated and the relevant references cited.

**RC1.2.** *A number of off the shelf data products are used uncritically. I note some lip service is paid to this in the final paragraph of the discussion, but that did not seem to influence the rationale of the analyses. There does not appear to be any thought given to the evaluation of the accuracy of these products, particularly with respect to their extreme values. In any process or empirical modelling context, the extreme values tend to be the hardest to accurately predict. This study seems to be based on linear relationships of the extreme values of modelled plant functional traits to the extreme values of modelled climate data. I think this results in regressions of speculative model noise and outliers against other model outliers.*

**Response:** We respectfully disagree with the assessment that we used off-the-shelf data products uncritically. We used data that has been published, trusting that the peer reviewer process made these datasets of quality for subsequent studies. Further, these same datasets have been used in additional studies, for example (Liu et al., 2023; Peng et al., 2023; Lin, Zhu, Wang, Zhang, & Yu, 2024). However, in a revised version we will provide further clarification on the quality of the data and the applications both in the methods and in the discussion.

You ask about the quality of the products with respect to the extreme values. While we agree with your perspective that indeed the extremes are those more difficult to predict and there is the potential that indeed the results could be sensitive to outliers, the data we used is globally available and was specifically cropped to the Amazon Basin. The range of values within the Amazon Basin does not represent the extremes of the global dataset. Furthermore, the quantile regression approach employed in our study does not simply map extreme values of independent variables to extreme values of dependent variables. Instead, at a given quantile, the model assigns varying weights to the corresponding quantiles of the dependent variable (Householder et al., 2024; ter Steege et al., 2023). This method, in fact, reduces susceptibility to extreme values compared to standard linear regression, enabling us to explore potential relationships that may exist at the extremes of the data distribution. In the revised version we will review the text to clarify the quality control and outlier analyses that the original authors of the datasets have conducted.

**RC1.3.** *I also must note that weather and climate data (which are intrinsically dynamic) are not typically called parameters (which are usually static in a modelling context), nor is it very compelling to frame a story around 'parameters'.*

**Response:** We agree, yet we found it in the literature, and will rephrase to "variables" for clarity.

**RC1.4.** *I question the logic of the assumed causality in the conclusions presented here. Why is it concluded that the plant functional traits will affect VPD and LST, and not vice-versa? In reality, this is at least partially circular. LST, or rather the departure of LST from Tair will reflect latent heat flux. Plant transpiration is primarily a function of stomatal conductance and VPD, amongst other things. But the manuscript does not delve into what modulates stomatal conductance, nor relate it (stomatal slope) to the modelled plant functional traits. Ultimately, I think this analysis is naively empirical and ignores decades of theory from ecophysiology and land-surface modelling. My impression is that this represents a data mining exercise (48 data combinations with three quantiles)*

**Response:** Thank you for your concern. We expect a feedback effect, i.e. plant functional traits affect VPD and LST and will be affected by these properties. In fact, plant traits enable the regulation of water exchanges between the canopy and the atmosphere. Traits like SLA, LDMC, LNC and LPC are all fundamental for photosynthesis and therefore, perhaps indirectly, will affect the regulation of atmospheric water demand (VPD) and LST (through latent heat). In the reviewed version, we will provide a conceptual diagram explaining this feedback.

However, we respectfully disagree with the comment that our analysis constitutes a data mining exercise. Our study aims to evaluate the effects of plant traits on a set of water cycle processes. We chose quantile regression because we want to examine if the slope of the relationship was steeper or flatter in different parts of the distribution (Cade & Noon, 2003; Antúnez, Wehenkel, Hernández-Díaz, & Garza-López, 2023; ter Steege et al., 2023; Householder et al., 2024).

**RC1.5.** *The data analysis paragraph is exceptionally short and does not provide sufficient detail for me to begin to understand what was done, nor how. Were these single or multiple regressions? This is far from reproducible.*

**Response:** Thank you for your assessment. We will expand the description. We state in the manuscript that "We conducted quantile regressions for each trait–water cycle parameter pair" (Line 173), which is explicitly indicative of a univariate approach. We will make it more explicit by explaining these were univariate quantile regressions in the revised manuscript.

**RC1.6.** *I did not understand the justification for relating these regressions to the sub-basins.*

**Response:** We included sub-basins in our analysis because we wanted to examine whether the relationships would be similar or different from those at the Basin scale. We explained this in the Results section, and we agree that these details could be incorporated into the "Study Area" subsection to more comprehensively justify our sub-basin analyses.

**RC1.7.** *The highest R2 value is 0.082, which is still quite low, nor does there appear to be any attempt at correcting for model overfitting. If one fits a large number of models to random data, I would expect some occasional high R2 values.*

**Response:** The R2 value of 0.082 that RC1 mentions corresponds to the Regulation of Soil Moisture Content, which we mentioned was not significant. We discussed the possible interpretation of this result in the discussion section. Additionally, we observed higher R2 values (between 0.142 - 0.278) in the Regulation of Evapotranspiration and Land Surface Temperature (Subsection 3.1.1-2 in

the manuscript).

**RC1.8.** *There are 24 appendix tables. This seems extreme. Who will read these and for what purpose? Their presence is not strongly justified in the main text. The only appendix table referenced in the main text is A1*

**Response:** The appendix tables are included to present the results of statistical analysis using quantile regression, as the main text focuses only on significant relationships. However, we can move these tables to the Supplementary Materials.

**RC1.9.** *Figure 2: All of these panels are lacking units, and these scalebars are not colorblind friendly. The numbers on LSTDAY range from 13750 to 15500. What does this correspond to? Average based on what time period? This is really lacking a lot of standard details.*

**Response:** Thank you for your comment. The omission of units in the figure was an oversight on our part, and we will correct this in the revision. We will also use color-blind friendly color palletes.

Regarding the time period for averaging, we provided this information in the manuscript. Specifically, we mentioned that the data were acquired "between 2001 and 2010 to match the timeframe of the trait data" (Line 145) and that we "calculated the 10-year mean and standard deviation for ET, PET, VPD, LSTday, LSTnight, and SM" (Line 162).

**RC1.10.** *Figure 3: I am at a loss for how to interpret this. Does every single row for each panel represent the R2 of a different quantile regression model?*

**Response:** We have explained in the figure caption that the bars represent the R2 values for the 5th, 50th, and 95th quantiles. Additionally, the corresponding quantiles, water cycle variables, and plant traits are clearly annotated in the figure. Given this, we are unsure what specifically needs to be changed in the figure. We would appreciate guidance on this.

**RC1.11.** *Figures 4, 5: Again, I am at a loss for how to interpret this. What do the numbers on the x-axis correspond to? Specifically what does standardised mean on the y-axis? Z-score transformed?*

**Response:** The plots are presented as a matrix, where the y-axis represents water cycle variables, and the x-axis represents plant traits, as our analysis was to assess whether water cycle variables were affected by plant traits. Additionally, in the legend we explain that the colors correspond to the regression lines for the different quantiles. If there is specific ambiguity regarding the axes or standardization, we would appreciate further clarification so we can make the figure clearer for all readers.

**References**

Antúnez, P., Wehenkel, C., Hernández-Díaz, J., & Garza-López, M. (2023). Quantile Regression as a Complementary Tool for Modelling Biological Data with High Variability. *Journal of Tropical Forest Science*, *35*(2), 130–140. Retrieved 2024-11-25, from `https://www.jstor.org/stable/48723350` (Publisher: Forest Research Institute Malaysia)

Cade, B. S., & Noon, B. R. (2003). A gentle introduction to quantile regression for ecologists. *Frontiers in Ecology and the Environment*, *1*(8), 412–420. Retrieved 2024-11-25, from https://onlinelibrary.wiley.com/doi/abs/10.1890/1540-9295%282003%29001%5B0412% 3AAGITQR%5D2.0.CO%3B2 (_eprint: https://onlinelibrary.wiley.com/doi/pdf/10.1890/1540-9295%282003%29001%5B0412%3AAGITQR%5D2.0.CO%3B2) doi: 10.1890/1540-9295(2003) 001[0412:AGITQR]2.0.CO;2

Householder, J. E., Wittmann, F., Schöngart, J., Piedade, M. T. F., Junk, W. J., Latrubesse, E. M., Quaresma, A. C., Demarchi, L. O., de S. Lobo, G., Aguiar, D. P. P. d., Assis, R. L., Lopes, A., Parolin, P., Leão do Amaral, I., Coelho, L. d. S., de Almeida Matos, F. D., Lima Filho, D. d. A., Salomão, R. P., Castilho, C. V., Guevara-Andino, J. E., Carim, M. d. J. V., Phillips, O. L., Cárdenas López, D., Magnusson, W. E., Sabatier, D., Revilla, J. D. C., Molino, J.-F., Irume, M. V., Martins, M. P., Guimarães, J. R. d. S., Ramos, J. F., Rodrigues, D. d. J., Bánki, O. S., Peres, C. A., Pitman, N. C. A., Hawes, J. E., Almeida, E. J., Barbosa, L. F., Cavalheiro, L., dos Santos, M. C. V., Luize, B. G., Novo, E. M. M. d. L., Núñez Vargas, P., Silva, T. S. F., Venticinque, E. M., Manzatto, A. G., Reis, N. F. C., Terborgh, J., Casula, K. R., Costa, F. R. C., Honorio Coronado, E. N., Monteagudo Mendoza, A., Montero, J. C., Feldpausch, T. R., Aymard C, G. A., Baraloto, C., Castaño Arboleda, N., Engel, J., Petronelli, P., Zartman, C. E., Killeen, T. J., Rincón, L. M., Marimon, B. S., Marimon-Junior, B. H., Schietti, J., Sousa, T. R., Vasquez, R., Mostacedo, B., Dantas do Amaral, D., Castellanos, H., Medeiros, M. B. d., Simon, M. F., Andrade, A., Camargo, J. L., Laurance, W. F., Laurance, S. G. W., Farias, E. d. S., Lopes, M. A., Magalhães, J. L. L., Mendonça Nascimento, H. E., Queiroz, H. L. d., Brienen, R., Stevenson, P. R., Araujo-Murakami, A., Baker, T. R., Cintra, B. B. L., Feitosa, Y. O., Mogollón, H. F., Noronha, J. C., Barbosa, F. R., de Sá Carpanedo, R., Duivenvoorden, J. F., Silman, M. R., Ferreira, L. V., Levis, C., Lozada, J. R., Comiskey, J. A., Draper, F. C., Toledo, J. J. d., Damasco, G., Dávila, N., García-Villacorta, R., Vicentini, A., Cornejo Valverde, F., Alonso, A., Arroyo, L., Dallmeier, F., Gomes, V. H. F., Jimenez, E. M., Neill, D., Peñuela Mora, M. C., Carvalho, F. A., Coelho de Souza, F., Feeley, K. J., Gribel, R., Pansonato, M. P., Ríos Paredes, M., Barlow, J., Berenguer, E., Dexter, K. G., Ferreira, J., Fine, P. V. A., Guedes, M. C., Huamantupa-Chuquimaco, I., Licona, J. C., Pennington, T., Villa Zegarra, B. E., Vos, V. A., Cerón, C., Fonty, , Henkel, T. W., Maas, P., Pos, E., Silveira, M., Stropp, J., Thomas, R., Daly, D., Milliken, W., Pardo Molina, G., Vieira, I. C. G., Albuquerque, B. W., Campelo, W., Emilio, T., Fuentes, A., Klitgaard, B., Marcelo Pena, J. L., Souza, P. F., Tello, J. S., Vriesendorp, C., Chave, J., Di Fiore, A., Hilário, R. R., Pereira, L. d. O., Phillips, J. F., Rivas-Torres, G., van Andel, T. R., von Hildebrand, P., Balee, W., Barbosa, E. M., Bonates, L. C. d. M., Doza, H. P. D., Gómez, R. Z., Gonzales, T., Gonzales, G. P. G., Hoffman, B., Junqueira, A. B., Malhi, Y., Miranda, I. P. d. A., Mozombite-Pinto, L. F., Prieto, A., Rudas, A., Ruschel, A. R., Silva, N., Vela, C. I. A., Zent, S., Zent, E. L., Cano, A., Carrero Márquez, Y. A., Correa, D. F., Costa, J. B. P., Flores, B. M., Galbraith, D., Holmgren, M., Kalamandeen, M., Nascimento, M. T., Oliveira, A. A., Ramirez-Angulo, H., Rocha, M., Scudeller, V. V., Sierra, R., Tirado, M., Umaña, M. N., van der Heijden, G., Vilanova Torre, E., Ahuite Reategui, M. A., Baider, C., Balslev, H., Cárdenas, S., Casas, L. F., Farfan-Rios, W., Ferreira, C., Linares-Palomino, R., Mendoza, C., Mesones, I., Parada, G. A., Torres-Lezama, A., Urrego Giraldo, L. E., Villarroel, D., Zagt, R., Alexiades, M. N., de Oliveira, E. A., Garcia-Cabrera, K., Hernandez, L., Palacios Cuenca, W., Pansini, S., Pauletto, D., Ramirez Arevalo, F., Sampaio, A. F., Valderrama Sandoval, E. H.,

Valenzuela Gamarra, L., & ter Steege, H. (2024, May). One sixth of Amazonian tree diversity is dependent on river floodplains. *Nature Ecology & Evolution*, *8*(5), 901–911. Retrieved 2024-11-25, from `https://www.nature.com/articles/s41559-024-02364-1` (Publisher: Nature Publishing Group) doi: 10.1038/s41559-024-02364-1

Jetz, W., Cavender-Bares, J., Pavlick, R., Schimel, D., Davis, F. W., Asner, G. P., Guralnick, R., Kattge, J., Latimer, A. M., Moorcroft, P., Schaepman, M. E., Schildhauer, M. P., Schneider, F. D., Schrodt, F., Stahl, U., & Ustin, S. L. (2016, March). Monitoring plant functional diversity from space. *Nature Plants*, *2*(3), 1–5. Retrieved 2024-12-20, from `https://www.nature.com/articles/nplants201624` (Publisher: Nature Publishing Group) doi: 10.1038/nplants.2016 .24

Lin, Q., Zhu, J., Wang, Q., Zhang, Q., & Yu, G. (2024). Patterns and drivers of atmospheric nitrogen deposition retention in global forests. *Global Change Biology*, *30*(7), e17410. Retrieved 2024-11-25, from `https://onlinelibrary.wiley.com/doi/abs/10.1111/gcb.17410` (_eprint: https://onlinelibrary.wiley.com/doi/pdf/10.1111/gcb.17410) doi: 10.1111/gcb.17410

Liu, Z., Chen, Z., Yu, G., Zhang, W., Zhang, T., & Han, L. (2023, February). The role of climate, vegetation, and soil factors on carbon fluxes in Chinese drylands. *Frontiers in Plant Science*, *14*. Retrieved 2024-11-25, from `https://www.frontiersin.org/journals/plant-science/articles/10.3389/fpls.2023.1060066/full` (Publisher: Frontiers) doi: 10.3389/fpls.2023 .1060066

Peng, Y., Prentice, I. C., Bloomfield, K. J., Campioli, M., Guo, Z., Sun, Y., Tian, D., Wang, X., Vicca, S., & Stocker, B. D. (2023). Global terrestrial nitrogen uptake and nitrogen use efficiency. *Journal of Ecology*, *111*(12), 2676–2693. Retrieved 2024-11-25, from `https://onlinelibrary.wiley.com/doi/abs/10.1111/1365-2745.14208` (_eprint: https://onlinelibrary.wiley.com/doi/pdf/10.1111/1365-2745.14208) doi: 10.1111/1365-2745 .14208

ter Steege, H., Pitman, N. C. A., do Amaral, I. L., de Souza Coelho, L., de Almeida Matos, F. D., de Andrade Lima Filho, D., Salomão, R. P., Wittmann, F., Castilho, C. V., Guevara, J. E., Veiga Carim, M. d. J., Phillips, O. L., Magnusson, W. E., Sabatier, D., Revilla, J. D. C., Molino, J.-F., Irume, M. V., Martins, M. P., da Silva Guimarães, J. R., Ramos, J. F., Bánki, O. S., Piedade, M. T. F., Cárdenas López, D., Rodrigues, D. d. J., Demarchi, L. O., Schöngart, J., Almeida, E. J., Barbosa, L. F., Cavalheiro, L., dos Santos, M. C. V., Luize, B. G., de Leão Novo, E. M. M., Vargas, P. N., Silva, T. S. F., Venticinque, E. M., Manzatto, A. G., Reis, N. F. C., Terborgh, J., Casula, K. R., Honorio Coronado, E. N., Monteagudo Mendoza, A., Montero, J. C., Costa, F. R. C., Feldpausch, T. R., Quaresma, A. C., Castaño Arboleda, N., Zartman, C. E., Killeen, T. J., Marimon, B. S., Marimon-Junior, B. H., Vasquez, R., Mostacedo, B., Assis, R. L., Baraloto, C., do Amaral, D. D., Engel, J., Petronelli, P., Castellanos, H., de Medeiros, M. B., Simon, M. F., Andrade, A., Camargo, J. L., Laurance, W. F., Laurance, S. G. W., Maniguaje Rincón, L., Schietti, J., Sousa, T. R., de Sousa Farias, E., Lopes, M. A., Magalhães, J. L. L., Nascimento, H. E. M., de Queiroz, H. L., Aymard C., G. A., Brienen, R., Stevenson, P. R., Araujo-Murakami, A., Baker, T. R., Cintra, B. B. L., Feitosa, Y. O., Mogollón, H. F., Duivenvoorden, J. F., Peres, C. A., Silman, M. R., Ferreira, L. V., Lozada, J. R., Comiskey, J. A., Draper, F. C., de Toledo, J. J., Damasco, G., García-Villacorta, R., Lopes, A., Vicentini, A., Cornejo Valverde, F., Alonso, A., Arroyo, L., Dallmeier, F., Gomes, V. H. F., Jimenez, E. M., Neill, D., Peñuela Mora, M. C., Noronha, J. C., de Aguiar, D. P. P.,

Barbosa, F. R., Bredin, Y. K., de Sá Carpanedo, R., Carvalho, F. A., de Souza, F. C., Feeley, K. J., Gribel, R., Haugaasen, T., Hawes, J. E., Pansonato, M. P., Ríos Paredes, M., Barlow, J., Berenguer, E., da Silva, I. B., Ferreira, M. J., Ferreira, J., Fine, P. V. A., Guedes, M. C., Levis, C., Licona, J. C., Villa Zegarra, B. E., Vos, V. A., Cerón, C., Durgante, F. M., Fonty, , Henkel, T. W., Householder, J. E., Huamantupa-Chuquimaco, I., Pos, E., Silveira, M., Stropp, J., Thomas, R., Daly, D., Dexter, K. G., Milliken, W., Molina, G. P., Pennington, T., Vieira, I. C. G., Weiss Albuquerque, B., Campelo, W., Fuentes, A., Klitgaard, B., Pena, J. L. M., Tello, J. S., Vriesendorp, C., Chave, J., Di Fiore, A., Hilário, R. R., de Oliveira Pereira, L., Phillips, J. F., Rivas-Torres, G., van Andel, T. R., von Hildebrand, P., Balee, W., Barbosa, E. M., de Matos Bonates, L. C., Dávila Doza, H. P., Zárate Gómez, R., Gonzales, T., Gallardo Gonzales, G. P., Hoffman, B., Junqueira, A. B., Malhi, Y., de Andrade Miranda, I. P., Pinto, L. F. M., Prieto, A., Rudas, A., Ruschel, A. R., Silva, N., Vela, C. I. A., Zent, E. L., Zent, S., Cano, A., Carrero Márquez, Y. A., Correa, D. F., Costa, J. B. P., Flores, B. M., Galbraith, D., Holmgren, M., Kalamandeen, M., Lobo, G., Torres Montenegro, L., Nascimento, M. T., Oliveira, A. A., Pombo, M. M., Ramirez-Angulo, H., Rocha, M., Scudeller, V. V., Sierra, R., Tirado, M., Umaña, M. N., van der Heijden, G., Vilanova Torre, E., Reategui, M. A. A., Baider, C., Balslev, H., Cárdenas, S., Casas, L. F., Endara, M. J., Farfan-Rios, W., Ferreira, C., Linares-Palomino, R., Mendoza, C., Mesones, I., Parada, G. A., Torres-Lezama, A., Urrego Giraldo, L. E., Villarroel, D., Zagt, R., Alexiades, M. N., de Oliveira, E. A., Garcia-Cabrera, K., Hernandez, L., Cuenca, W. P., Pansini, S., Pauletto, D., Ramirez Arevalo, F., Sampaio, A. F., Valderrama Sandoval, E. H., Gamarra, L. V., Levesley, A., Pickavance, G., & Melgaço, K. (2023, November). Mapping density, diversity and species-richness of the Amazon tree flora. *Communications Biology*, *6*(1), 1–14. Retrieved 2024-11-25, from `https://www.nature.com/articles/s42003-023-05514-6` (Publisher: Nature Publishing Group) doi: 10.1038/s42003-023-05514-6

---

## Author Comment (AC2)

**Responses to comments from RC2**

**Manuscript: Effects of plant traits on the regulation of water cycle processes in the Amazon Basin**

**Kien Nguyen and Maria J. Santos**

Dear editor,

With this letter, we provide our reply to the comments provided by RC2 to our submitted manuscript entitled *"Effects of plant traits on the regulation of water cycle processes in the Amazon Basin"*. We appreciate the time that RC2 took to review our work and have seriously considered and appreciated all the comments by the reviewer. We detail in the following our plan to address the comments in a revised version.

**General Comments**

*This study aims to investigate the effects of plant traits on regulating water cycle processes in the Amazon basin. However, the author refers to 'water process parameters': yet, ET, PET, VPD, LST, and SM are not strictly 'parameters'. Additionally, the paper lacks clear organization; the Discussion section contains significant overlap with the Introduction.*

**Response:** Thank you for your comments on our work. We will revise the terminology in the manuscript. Additionally, we will restructure the Discussion and Introduction sections to enhance relevance and reduce overlap.

**RC2.1.** *Abstract: could the author specify how much of the variation in ET, PET, and other variables can be explained by plant traits?*

**Response:** The explained variance relating to ET, PET and other variables could be found in the Results section (Figure 3).

**RC2.2.** *Clarity of research focus - \*... to examine how plant traits respond to parameters related to regulation of atmospheric water content ... \* This is unclear. It suggests an analysis of how external processes influence plant traits, while the rest of the paper seems to focus on the opposite direction - how plants regulate water cycle processes. This should be clarified.*

**Response:** We aim to link plant traits to water process variables, treating water process variables as response variables and plant traits as explanatory variables. We apologize for the lack of clarity in the manuscript and will clarify this in the revised version.

**RC2.3.** *We found that SLA had the strongest relationships with parameters ... * The author should specify the strength of these relationships - e.g. correlation coefficients.

**Response:** We agree and will include the corresponding correlation coefficients in the revised manuscript.

**RC2.4.** *Do Figure 4 and Figure 5 provide new insights, or do they largely repeat the information already presented in Figure 3?*

**Response:** The information in Figures 4 and 5 differs from that in Figure 3. Figures 4 and 5 illustrate the direction and slope of the quantile regression, whereas Figure 3 presents the explained variance.

**RC2.5.** *While the paper highlights the importance of plant trait diversity and spatial heterogeneity, it assumes a unidirectional relationship between plant traits and water cycle processes. In reality, these interactions are bidirectional—plant traits regulate transpiration, leaf temperature, and soil moisture dynamics, but environmental conditions also influence plant traits through acclimation or adaptation. The paper focuses on analyzing forward relationships, but this approach alone may not be sufficient. I strongly suggest employing causality inference methods to examine these relationships in a more robust manner.*

**Response:** We agree with the argument raised by RC2. We will reconsider our analysis approach in a revised version of the manuscript.

**RC2.6.** *L260-263: The author should provide stronger evidence to support their claims rather than attributing unexpected results solely to data limitations. Simply stating that the data are insufficient is not a compelling explanation.*

**Response:** We appreciate this suggestion and agree with the point raised by RC2. In the revised version, we will provide stronger evidence to support our claims and explore alternative explanations beyond data limitations.

**RC2.7.** *Establishing a direct link between LPC and VPD seems problematic, as VPD is primarily a meteorological factor. While vegetation dynamics can influence background climate conditions, this relationship should be better justified.*

**Response:** We appreciate this feedback and will revise the manuscript to clarify the relationship between LPC and VPD. We will provide a more detailed justification, including relevant literature and additional analysis, to explore the potential influence of vegetation on background climate conditions.

**RC2.8.** *Section 4.3: Why do plant traits significantly relate to transpiration but not to soil moisture? A more detailed discussion or additional analysis would be helpful.*

**Response:** We will revise the manuscript to provide a more detailed discussion on the relationship between plant traits and soil moisture.

**RC2.9.** *Figure 1: I suggest using illustrations (e.g., arrows, plus/minus symbols) to better depict the proposed mechanisms while keeping explanatory text within the main text. Additionally, the left-side arrow should be explicitly labeled (e.g., transpiration and soil rehydration) for clarity.*
*Figure 2: the unit for each panel is missing. What is the unit of LST? And the tick labels on both axes are wrong. You may either leave them blank or use the correct latitude and longitude like panel (a). The caption for Table 1 is too brief.*

**Response:** Thank you for your comments and suggestions. We will revise the figures and the table accordingly.